# Multiscale architecture design of 3D printed biodegradable Zn-based porous scaffolds for immunomodulatory osteogenesis

Shuang Li[1,8], Hongtao Yang [1,2,8] ✉, Xinhua Qu[3,8], Yu Qin[2], Aobo Liu[4], Guo Bao[5], He Huang[6], Chaoyang Sun[1], Jiabao Dai[4], Junlong Tan [1], Jiahui Shi[2], Yan Guan[7], Wei Pan[7], Xuenan Gu[1], Bo Jia [4], Peng Wen [4] ✉, Xiaogang Wang [1] ✉ & Yufeng Zheng [2] ✉

Reconciling the dilemma between rapid degradation and overdose toxicity is challenging in biodegradable materials when shifting from bulk to porous materials. Here, we achieve significant bone ingrowth into Zn-based porous scaffolds with 90% porosity via osteoinmunomodulation. At microscale, an alloy incorporating 0.8 wt% Li is employed to create a eutectoid lamellar structure featuring the $LiZn_4$ and Zn phases. This microstructure optimally balances high strength with immunomodulation effects. At mesoscale, surface pattern with nanoscale roughness facilitates filopodia formation and macrophage spreading. At macroscale, the isotropic minimal surface G unit exhibits a proper degradation rate with more uniform feature compared to the anisotropic BCC unit. In vivo, the G scaffold demonstrates a heightened efficiency in promoting macrophage polarization toward an anti-inflammatory phenotype, subsequently leading to significantly elevated osteogenic markers, increased collagen deposition, and enhanced new bone formation. In vitro, transcriptomic analysis reveals the activation of JAK/STAT pathways in macrophages via up regulating the expression of *Il-4*, *Il-10*, subsequently promoting osteogenesis.

Metallic zinc (Zn) has been widely used for the unique electrochemical and structural properties in human history since its first discovery in 18th century. Meanwhile, Zn is the second abundant transition metal in humans, functioning as structural or enzymatic cofactors for roughly 10% of the proteome[1]. In 21st century, the marriage of metallic Zn with Zn biology gives birth to Zn-based biomaterials as new methodology to address the unsolved clinical challenges[2,3]. In the field of orthopedics, Zn-based biodegradable materials, characterized by excellent mechanical properties, osteogenic and antibacterial bioactivities, are under wide investigations for potential applications as screw and plate systems[4,5], intramedullary needles[6], bone grafts[7,8], guided bone regeneration membranes[9], etc. The first clinical trial using zinc alloy implants was reported for the treatment of craniofacial fractures in China in 2020 (ChiCTR registration number: ChiCTR2100051050).

[1]School of Engineering Medicine, School of Biological Science and Medical Engineering, Beihang University, 100191 Beijing, China. [2]School of Materials Science and Engineering, Peking University, 100871 Beijing, China. [3]Department of Bone and Joint Surgery, Department of Orthopedics, Renji Hospital, Shanghai Jiao Tong University School of Medicine, 200001 Shanghai, China. [4]Department of Mechanical Engineering, Tsinghua University, 100084 Beijing, China. [5]Department of Reproduction and Physiology National Research Institute for Family Planning, 100081 Beijing, China. [6]School of Materials Science and Engineering, Zhengzhou University, 450003 Zhengzhou, China. [7]College of Chemistry and Molecular Engineering, Peking University, 100871 Beijing, China. [8]These authors contributed equally: Shuang Li, Hongtao Yang, Xinhua Qu. ✉e-mail: yang276070@buaa.edu.cn; wenpeng@tsinghua.edu.cn; xiaogangwang@buaa.edu.cn; yfzheng@pku.edu.cn

These in vivo and clinical studies reveal a discouraging mismatch that the degradation of bulk Zn-based implants may typically require a decade or more, while the physiological cycle of bone repair spans only 3−6 months. In the past three years, the 3D printing technology based on novel biodegradable Zn alloys has rapidly advanced, aiming to reduce material consumption while maintaining sufficient mechanical performance through innovative structural designs[10–12]. However, overdose-induced Zn toxicity caused by accelerated degradation is a critical challenge in Zn-based porous scaffolds due to the increased porosity and specific surface area in comparison to bulk materials[13]. Understanding the impact of porous scaffold structure design on degradation dynamics of scaffold materials and corresponding biofunctions are crucial for the success of bone tissue repair.

As foreign bodies, biomaterials will immediately be signaled by the immune system, and triggers a cascade of inflammatory reactions, as part of tissue repair[14]. Biomaterials with osteoimmunomodulatory properties can positively modulate immune cells behavior and promote favorable tissue responses during bone regeneration. Numerous methods have been utilized to alter the interaction with immune cells, including the adjustment of chemical or structural properties and the integration of bioactive substances[15]. The vital role of immune cells in regulating the function of bone cells makes the paradigm shift of bone biomaterial design to osteoimmunomodulation[16]. Macrophages play a crucial role in bone tissue regeneration by regulating immune response, promoting angiogenesis, and modulating the activity of osteoblasts. Their close interaction with bone cells influences bone remodeling and healing. Osteal macrophages (OsteoMacs) represent a distinct subset of macrophages found within skeletal structures. Intriguing discoveries from foundational studies have highlighted their significant contributions to bone physiology, showcasing their pivotal involvement in both bone formation and remodeling processes[17,18]. Biodegradable metals start an electrochemical reaction upon contact with body fluids, thereby activating the innate immune system within hours. Macrophages are among the first arrivals to interact with implants and mediate the host foreign body response. For biodegradable Zn-based porous scaffolds, the interplay between material and immune cells and their mediated tissue repair is performed at multiple scales. At the micro level, divalent Zn ions are transported into macrophages via SLC30/SLC39 family and regulated by metallothioneins to mediate Zn homeostasis[19]. The intracellular Zn determines the cell fate of macrophages including viability, phagocytosis capability, polarization, and inflammatory signaling[20]. At the meso level, the abundant surface area of porous materials with topographical cues can serve as a natural platform for regulating macrophage behavior. $TiO_2$ honeycombs with 90 nm patterns promote the formation of filopodia in macrophages and activates the RhoA/ROCK signaling pathway, which further regulate macrophage polarization and cytokine secretion[21]. At the macro level, the pore geometry can influence collective cell behavior and the spatiotemporal characteristics of material degradation[22]. For example, multicellular spatiotemporal organization of pre-osteoblasts indicates cells prefer regions with at least one negative principal curvature[23]. The degradation of Zn alloy scaffolds increases positively with porosity while negatively with pore size[10]. Therefore, multiscale architecture design for biodegradable Zn-based porous scaffolds may successfully orchestrate the immune cell responses and the subsequent bone regeneration.

Here, we fabricated Zn-based alloy scaffolds with 90% porosity, marking the highest reported porosity to date. The specific surface area of the scaffold is 10 times greater than its bulk counterpart. To mitigate the risk of zinc-induced overdose toxicity, we selected high strength binary Zn-Li alloy system and investigated the impacts of degradation on local physicochemical factors. Then, we examined the co-release of Zn and Li ions on polarization and inflammatory cytokine expression of RAW 264.7 macrophages. The Zn-Li alloy with the optimal comprehensive performance was used to fabricate porous scaffolds with a laser powder bed fusion technique. Ultrasound treatment, acid etching, and electrochemical polishing were developed to create micro patterns with distinct morphologies on the scaffold surface. The surface roughness and cell morphology on 2D surface and 3D structure were investigated. The optimal combination between material composition and surface pattern was further used to create Zn-based porous scaffolds with a biomimetic minimal surface geometry (gyroid), using a traditional body centered cubic (BCC) lattice pore unit as comparison. The temporal relationships among structure-degradation-tissue regeneration focusing on osteoimmunomodulation are systematically examined at multiple levels including genes, cells and tissues.

## Results

### Composition design of Zn-based alloys

A superior mechanical property with proper surface corrosion and ion release behavior are the major considerations for the compositional design of biodegradable Zn-based scaffold with osteoimmunomodulation capability. For Zn-Li alloys, there is a trade-off in the correlation between strength, plasticity, and the addition of Li elements (Fig. 1A and Table S1). Alloying with Li increases the strength while decreases the plasticity of Zn-Li alloys. The optimal combination of strength and plasticity lies in Zn-0.8Li alloys. A lamellar eutectoid microstructure composed of Zn and $LiZn_4$ with interlamellar spacing at ~200−300 nm appears predominantly in Zn-0.8Li alloy (Fig. 1B). For electrochemical behavior (Fig. 1C), the surface impedance of materials increases greatly with Li contents when immersing in simulated body fluid (SBF). After 24 h immersion in SBF, zinc oxide (ZnO) and lithium carbonate ($Li_2CO_3$) are the major corrosion products detected by X-ray photoelectron spectroscopy (XPS) in Zn-Li alloys (Fig. 1D and Table S2). Surface potential distribution (Fig. 1E) in Zn-0.2Li and Zn-0.8Li alloys is similar, which is uniform when immersing in SBF. But the surface pH value in Zn-0.8Li alloy (9.2−9.3) is much higher than that of Zn-0.2Li alloy (7.5-7.6). Additionally, Li partially substitutes for Zn release during culture with cell medium (Fig. 1F), achieving a Zn:Li ratio of approximately 4:1 in the Zn-0.8Li alloy (Zn concentration: 17.1 μg/mL, Li concentration: 3.9 μg/mL). To evaluate the impact of released Zn and Li ions on immune cells, RAW 264.7 cells are co-cultured with material extracts for 48 h. The material extracts are 5-time diluted to prevent any cytotoxicity. As a result, more CD206 and less iNOS expression are found in Zn-Li alloy groups compared to the control and pure Zn groups (Figs. 1G and S1), especially in Zn-0.8Li groups. Meanwhile, the RNA expression of immunomodulatory cytokines like *Il-4*, *Il-10*, and *Arg1* are peaked at Zn-0.8Li alloy group while it inhibits the RNA expression of pro-inflammatory cytokines like *Tnf*-α, *iNos*, and *Il-1β* (Fig. 1H). As a result, Zn alloy with 0.8 wt.% addition of Li exhibits the optimal mechanical performance, proper and uniform electrochemical corrosion behavior. Most importantly, the co-release of Zn and Li ions from Zn-0.8Li alloy promotes the polarization of non-activated macrophages (M0) to macrophages with a pro-regenerative phenotype (M2) and stimulates the expression of immunomodulatory cytokines most efficiently.

### Surface morphology and properties of Zn-0.8Li scaffold

Subsequently, Zn-0.8Li alloy is atomized into powder particles and printed into porous scaffolds with a L-PBF technique. Unmelted powder is a common problem to affect many aspects of the performance, especially the biofunctions of 3D printed scaffolds after laser fusion. Ultrasonic treatment, acid etching and electrochemical (EC) polishing are adopted here to improve the surface quality and modulate the behavior of RAW264.7 cells. After ultrasonic treatment, the scaffold is still covered with unmelted powders and a dense oxide layer (Fig. 2A). Acid etching removes some of the loosely attached powders and exposes the micro patterns derived from the Zn-Li alloy microstructure. The AFM image displays orderly protrusions in acid etched

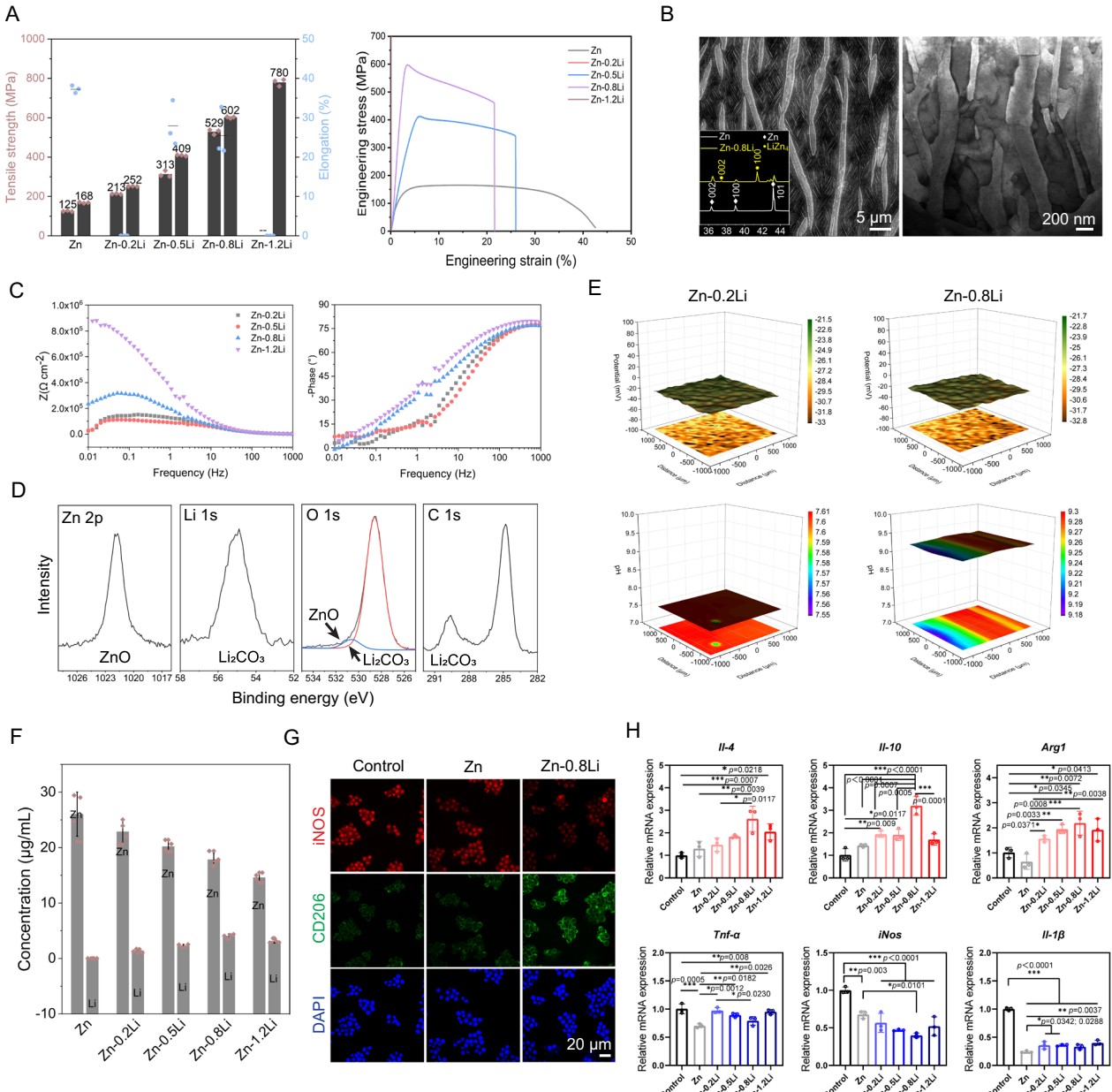

**Fig. 1 | Composition design of Zn-based alloys for biodegradable bone scaffolds. A** Mechanical property of Zn-Li alloys (*n* = 3, independent experiments). Data are presented as mean ± standard deviation. **B** Microstructure of Zn-0.8Li alloy with scanning electron microscope (SEM) and transmission electron microscope (TEM) images. The insert is the X-ray diffraction (XRD) pattern of pure Zn and Zn-0.8Li alloy. **C** Bode plots of Zn-Li alloys after immersing in SBF for 24 h. **D** X-ray photoelectron spectroscopy (XPS) measurement of spectrums of corrosion products after immersion. **E** Scanning vibrating electrode technique (SVET) monitoring of potential and pH distribution on Zn-Li alloy surface at 24 h after immersion.

**F** Concentrations of Zn and Li in extracts of Zn-Li alloys (*n* = 4, independent experiments). Data are presented as mean ± standard deviation.
**G** Immunofluorescence staining of iNOS, CD206, and DAPI of RAW 264.7 cells after co-culture with material extracts for 48 h. **H** RNA expression of cytokines including *Il-4*, *Il-10*, *Arg1*, *Tnf*-α, *iNos*, and *Il-1β* from cells cultured with material extracts (*n* = 3, independent experiments). Data are presented as mean ± standard deviation. *P*-values are calculated using one-way ANOVA with Tukey's post hoc test, **p* < 0.05, ***p* < 0.01, ****p* < 0.005. Each image was acquired independently three times, with similar results (**B**, **G**). Source data are provided as a Source Data file.

surface, and the Ra is about 79 nm (Fig. 2B). Further after EC polishing, the scaffold strut is smooth without attached powders. The EC polished surface shows a coarser protruding structure (Ra = 114 nm) with a wavy-like morphology compared to the acid-etched surface. RAW264.7 cells are seeded on the scaffold to see the interaction between cells and surface patterns (Figs. 2C and S2). Cells are in round and wrinkled morphology, and gather into clusters when attaching the ultrasounded or acid-etched surface. The unmelted powders seem to prevent cells from adhesion and spreading. In contrast, pseudopodium of cells anchors on the protruded patterns of EC polished

surface, and cell spreading area increases significantly compared to other groups. Nyquist and bode plots show that acid etching activates the scaffold surface with smaller impedance compared to the as-printed surface while EC polishing repassivates the surface to some extent (Fig. 2D).

## Mechanical and corrosion properties of Zn-Li porous scaffolds
Biomimic triple periodic minimal surface Gyroid (G) structure and traditional body-centered cubic (BCC) structure are selected as comparisons to study the impact of structures on mechanical property,

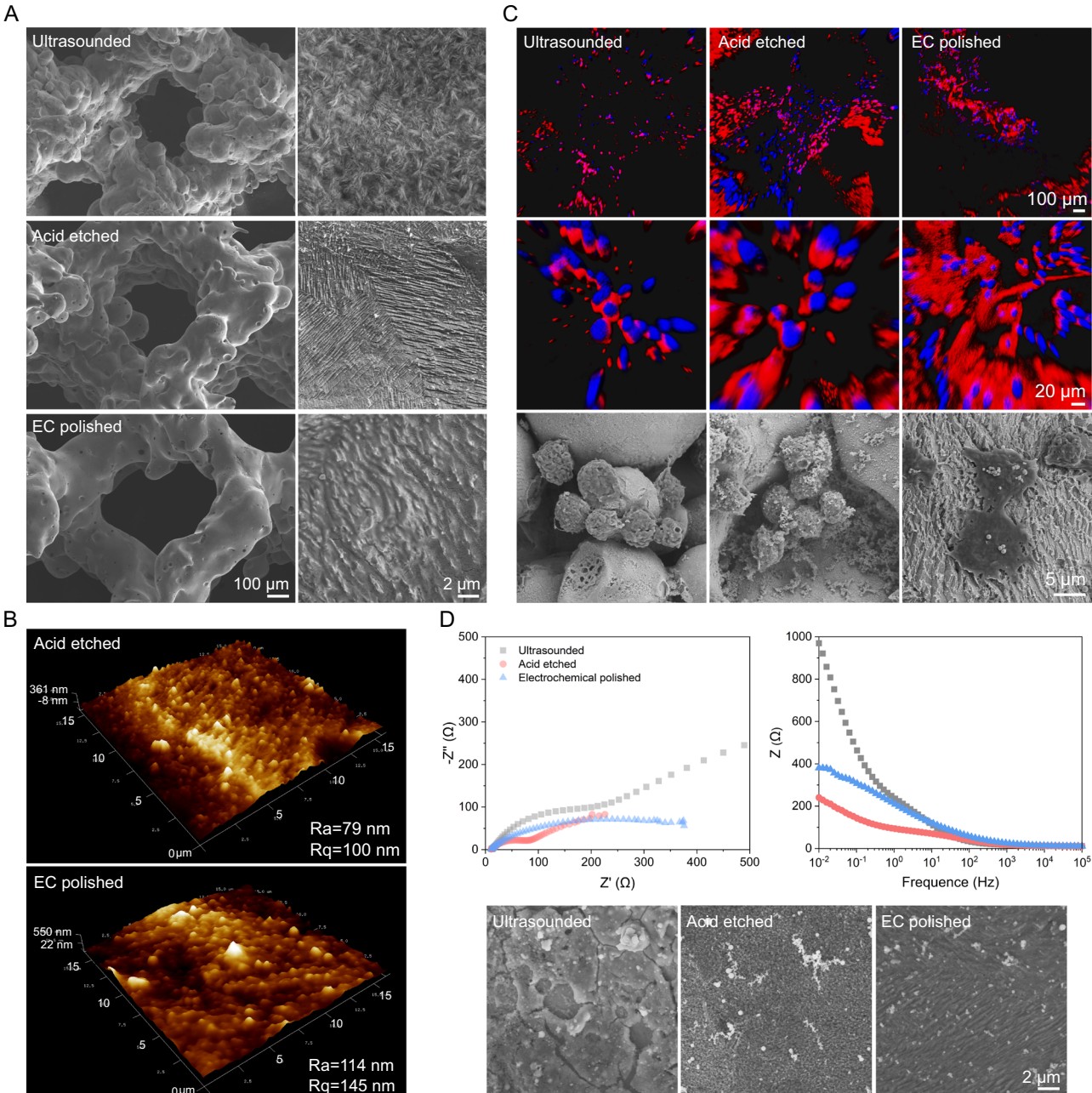

**Fig. 2 | Surface morphology and properties of Zn-0.8Li scaffold.**
**A** Representative SEM images of unit morphologies and surface patterns of 3D printed scaffolds after ultrasonic treatment, acid etching, and electrochemical (EC) polishing. **B** Atomic force microscope (AFM) images of surface patterns after acid etching, and EC polishing. Ra: absolute value roughness, Rq: root mean squared roughness. **C** F-actin (red)/DAPI (blue) staining of RAW264.7 cells and SEM images after 6 h attachment on different surfaces. **D** Nyquist and bode plots of different surface patterns and corresponding SEM morphologies after 24 h immersion. Each image was acquired independently three times, with similar results (**A**, **C**, **D**). Source data are provided as a Source Data file.

degradation behavior, and following biocompatibility and biofunctions. The porosity of scaffolds ranges from 86% to 90%, and their specific surface areas are 10 times more than the bulk control (Table S3). Figure 3A demonstrates the uniaxial compression behavior of Zn-Li porous scaffolds with BCC and G pore units. Additive manufacture and geometry design endow great freedom to manipulate the mechanical perform of the same material. The bulk Zn-Li sample has a compressive yield stress of 391.74 MPa, while adding 80% porosity reduces the material's compressive strength by 97%. When it comes to different pore units, the G structure shows higher strength and similar modulus compared to the BCC structure. In stress-strain curves, the BCC scaffold reaches its peak stress at ~2% strain followed by a

mechanical collapse. This is proved by the fracture morphology that cracks are found at intersection joints after its failure (around 10% strain), and sever deformation with more cracks appear thereafter. The stress of G structure shows a relative stable platform, and has higher resistance to crack propagation as cracks appears at intersection joints after 40% of strain. The G structure presents more uniform deformation behavior during compression.

Electrochemical tests (Fig. 3B) are performed to understand the impact of pore units on the early corrosion behavior of Zn-Li scaffolds. In the anode polarization curve with over potential <0.2 V, passivation platform appears in the G sample while BCC sample shows active dissolution feature. The corrosion current density of sample with G

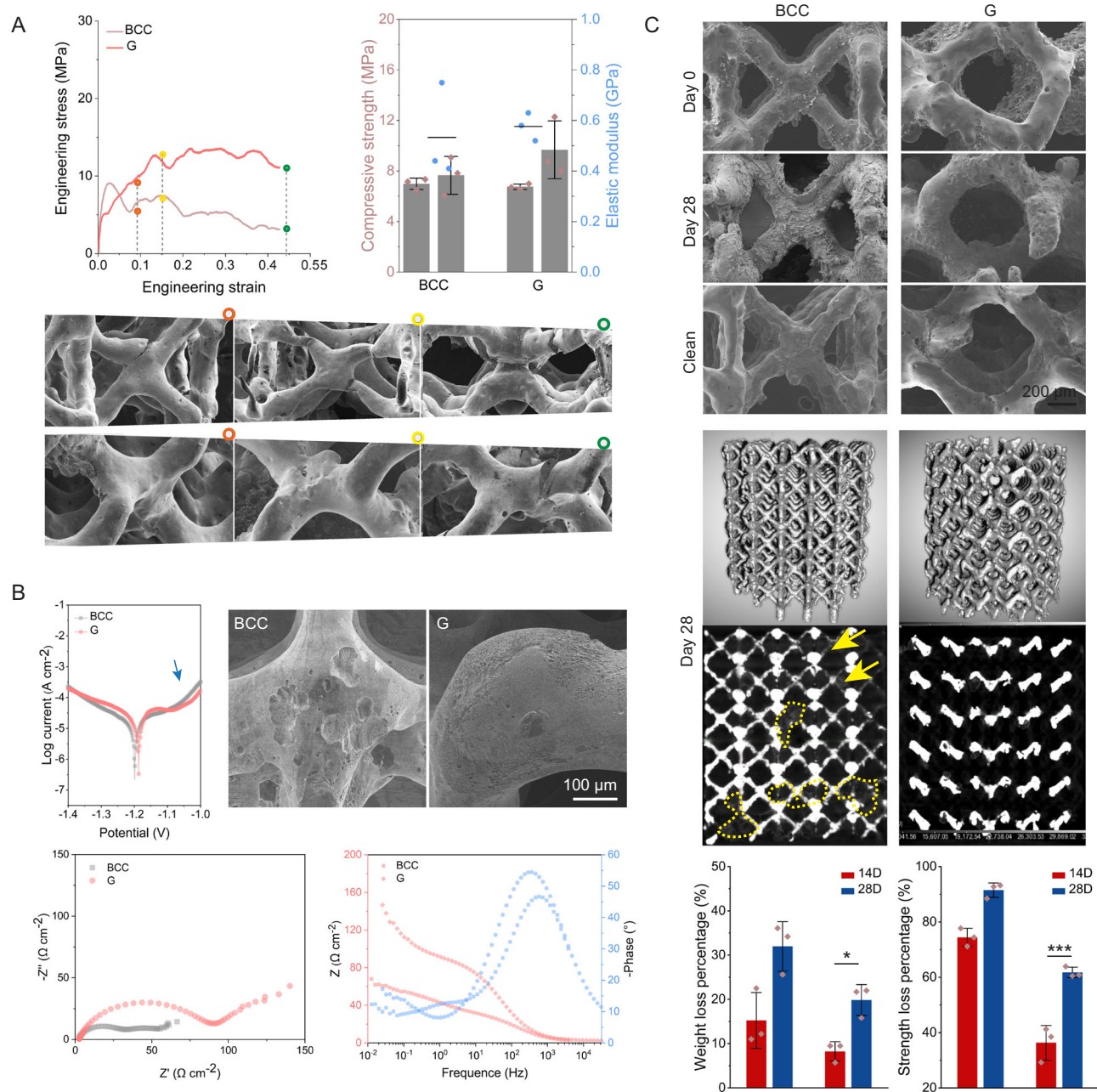

**Fig. 3 | Mechanical and corrosion properties of Zn-Li porous scaffolds with BCC and G pore units. A** Compressive strength (yield strength and ultimate strength) and modulus (n = 3, independent experiments). Data are presented as mean ± standard deviation. **B** Potentiodynamic polarization (PDP) curves and corrosion morphology of PDP samples after removal of corrosion products. Nyquist plots and Bode plots. **C** Corrosion morphology of BCC and G units at Day 0 and Day 28 after immersion in SBF. Clean indicates the morphology of units after removal of corrosion products. The 3D and 2D morphologies of samples at Day 28 are scanned by CT. Yellow arrows indicate corroded struts, yellow dashed lines indicate corrosion products. Weight loss percentage and compressive strength loss of Zn-Li porous scaffolds at Day 14 and 28 (n = 3, Independent experiments). Data are presented as mean ± standard deviation. P-values are calculated using one-way ANOVA with Tukey's post hoc test, *p < 0.05, **p < 0.01, ***p < 0.005. Each image was acquired independently three times, with similar results (**A**–**C**). Source data are provided as a Source Data file.

unit is slightly higher than that of the BCC unit. But the corrosion morphology in BCC sample manifests a non-uniform mode with large and deep pits concentrated in the connection region of struts. In contrast, small and shallow pits distribute homogeneously in the curved struts of the G sample. As shown in the nyquist and bode plots, the major difference between two units in impedance shows up in medium to low frequency. The charge transfer resistance in G scaffold is much high than that of the BCC scaffold. In low frequency region, mass transfer process seems be more dominant in the BCC scaffold than the G scaffold.

Samples are immersed in a specialized chamber with dynamic perfusion of simulated body fluid for 28 days to understand the impact of pore unit on the long-term corrosion behavior of Zn-Li porous scaffolds (Fig. 3C). The SEM morphology tells that all scaffolds degrade continuously as corrosion products accumulate on struts over time, and the struts are thinner after immersion. For BCC unit, some of the pores are sealed by corrosion products, and the metallic struts are severely corroded at 28 days as shown in the 2D CT image. In comparison, less corrosion products are found in scaffolds with the G unit, and most of the metallic struts stay intact at 28 days. The weight loss percentage of

BCC scaffold doubles compared to the G scaffold. As a result, 90% of the compressive strength of BCC scaffold loses while 40% of the mechanical integrity in G scaffold maintains after 28 days' immersion. Therefore, scaffolds with the G unit are more capable to maintain their structural and mechanical integrity compared to the BCC unit in the long-term corrosion test. The diffusion behavior of ions in different pore units are evaluated as shown in Fig. S3. The diffusion coefficient of Zn ions decreases over time. The BCC unit displays a strong anisotropy in terms of diffusion coefficient compared to the G unit.

## In vitro biocompatibility evaluation

RAW264.7 and MC3T3-E1 are two major effector cells during bone defect repair process. To evaluate the biocompatibility effects of Zn-Li porous scaffolds in vitro, the impact of unit geometry on adhesion and proliferation of RAW264.7 and MC3T3-E1 are presented in Fig. 4. Live/ dead staining shows that the majority of RAW264.7 and MC3T3-E1 cells survive on both scaffolds for 48 h. Statistical analyses reveal a three-fold increase in the adhesion of living cells to the surface of the G unit when compared to the BCC unit. Moreover, the number of cells adhering to the surface of the G scaffold per square millimeter is significantly higher than that of the BCC. SEM images demonstrate a healthy and spreading cell morphology with cell interconnections. Further analyses are performed by imaging the fluorescence of F-actin cytoskeleton in RAW264.7 stained with phalloidin at 48 h. Specifically, as indicated by the arrows, RAW264.7 cells on G unit have extensive F-actin-enriched filopodia extended radially from nuclei, with a much longer longitudinal stretch compared to cells on the BCC unit. As for MC3T3-E1, the number of cells on G scaffold is slightly higher than on BCC scaffold, but there is no statistical difference in the number of adherent cells per square millimeter of surface. MC3T3-E1 adhere tightly to the surface of porous scaffolds and spread out, cells elongate and interact with surrounding cells, present stretched cytoskeleton.

## In vitro macrophage cytokines modulate osteogenic differentiation of MC3T3-E1

To examine the impact of degradation product of Zn-0.8Li scaffolds on immune microenvironment, and the subsequent influence on bone formation, we investigated the impact of scaffold extracts on macrophage polarization, and the further influence of macrophage cytokines on osteogenic differentiation behavior of MC3T3-E1 pre-osteoblasts. The cell morphology of M0-type macrophages transformed from the typical round and spherical shape to elongated spindle-shaped cells, which is a characteristic morphological feature of M2-type macrophages after culturing in scaffold extracts for 48 h. Additionally, the nucleus to cytoskeleton area ratio is lower in G scaffold treated macrophages compared to those treated with BCC scaffold, indicating that G scaffold maximized cytoskeleton stretching (Fig. 5A). To determine the polarization status of macrophages, immunofluorescence staining is performed for the M1-type macrophage marker iNOS and the M2-type macrophage marker CD206. Both BCC and G scaffold groups significantly upregulated CD206 expression while inhibiting iNOS expression (Fig. 5B). Furthermore, to assess the secreted cytokines involved in macrophage polarization, mRNA levels of M1 pro-inflammatory factors (*iNos*, *Il-1β*, *Tnf-α*) and M2 anti-inflammatory factors (*Il-4*, *Il-10*, *Arg1*) are detected in RAW 264.7 cells using qRT-PCR analysis. As shown in Fig. 5C, gene expression associated with an M1 phenotype is down-regulated while gene expression related to an M2 phenotype is up-regulated in both BCC and G scaffold groups compared to the control group. G scaffold exhibits the lowest expression of *Tnf-α* and highest expression of *Il-4*, *Il-10* and *Arg1*.

To explore biomolecular mechanisms of macrophage polarization induced by the G scaffolds, transcriptomic analysis is performed to analyze signaling pathway differences in macrophages between the Zn and control groups (Fig. 5D). The volcano plot shows 287 up-regulated and 58 down-regulated genes, which indicates extensive gene expression differences. The results in the heat map display the

top-ranked 14 difference genes and depict fold changes of genes expression in G scaffold vs Control. The expression levels of M2-related genes (*Vegfa* and *Arg1*) are significantly enhanced in the G scaffold. Performing a Go analysis on all differentially expressed genes reveals enrichment in the regulation of cell proliferation, cell adhesion, cytoskeleton organization, and immune response. These processes induce M2 polarization through morphology and polarized growth, facilitated by G scaffold. Next, KEGG analysis is conducted to explore the underlying signaling pathways. We observe downregulation of pathways related to M1 macrophage activation, such as MAPK. Conversely, there is an upregulation of JAK-STAT pathways, consistent with M2 macrophage polarization (Fig. S4).

To investigate the impact of macrophage polarization on osteogenic differentiation, MC3T3-E1 cells were cultured in conditioned medium (CM) containing osteogenic components. After 7 days, ALP staining and activity results demonstrate significantly higher expression of ALP in both BCC and G scaffold groups compared to the control group. Particularly, the G scaffold group exhibits a remarkable increase in ALP positive expression. Similarly, at 14 days, ARS staining for calcium binding proteins in the mineralized matrix reveals that the size and quantity of calcium nodules are higher in the G scaffold group than in the BCC group, indicating more efficient calcium deposition by MC3T3-E1 cells (Fig. 5E, F). qRT-PCR analysis is employed to detect mRNA relative expression levels of *Alp*, *Opg*, *Opn*, and *Col1a1*. As shown in Fig. 5G, compared to the control group, both BCC and G groups show increased expression of osteogenesis-related genes. Notably, induction by the G scaffold results in the highest gene expression compared to BCC. Additionally, immunofluorescence staining for RUNX2 and OSX further confirms that the G scaffold group is most effective at promoting osteogenic differentiation (Fig. S5).

## In vivo degradation behavior

X-ray fluorescence imaging spectrometer (XRF), scanning electron microscope (SEM), and micro-CT are used to visualize the biodegradation of Zn-Li porous scaffolds in 2D and 3D over time. In general, all porous scaffolds exhibit significant macroscopic degradation after 3-month implantation compared to the bulk implant (Fig. S6). At 3 days, scaffold struts with intact contour are clearly seen in the bone defeat region (Fig. 6A). But XRF already detects the signal of Zn (navy blue) that distributes in the pore area and at the edge of the defeat region, indicating the early biodegradation of the scaffolds (Fig. 6B). At 1 month, severe degradation happens in the BCC scaffold as some of the pores are filled with degradation products completely. In contrast, the G scaffold degrade in a more uniform feature. Meanwhile, signal of Ca appears in the pore area of the G scaffold, which means the formation of mineralized matrix. At 3 months, the BCC scaffold shows significant but non-uniform degradation as some of the struts are totally degraded but others are not. This phenomenon can be clearly seen in the 3D reconstructed in vivo samples which shows a collapse morphology. The degradation mode of the G scaffold also changes from a homogeneous one to a localized one, but the metallic part of the scaffolds remains complete. More importantly, large amounts of new bone tissue grow into the interconnected pore area in the G scaffold as shown in SEM and XRF images. However, the BCC scaffold is still in the mineralized matrix deposition stage. Therefore, Zn-Li scaffolds with the G unit show a better match between material biodegradation and bone regeneration compared to the BBC unit.

## In vivo inflammatory response and early osteogenesis

Figure 7A shows the inflammatory response process of the BCC and G porous scaffold at the implant-tissue interface at 3 days and 1 month. Immunohistochemical staining exhibits iNOS (M1 phenotypic macrophages marker) and CD163 (M2 phenotypic macrophages marker) expression level in rat femoral tissue. In general, the implant-tissue interface mainly consisted of pro-inflammatory iNOS+ M1 macrophages

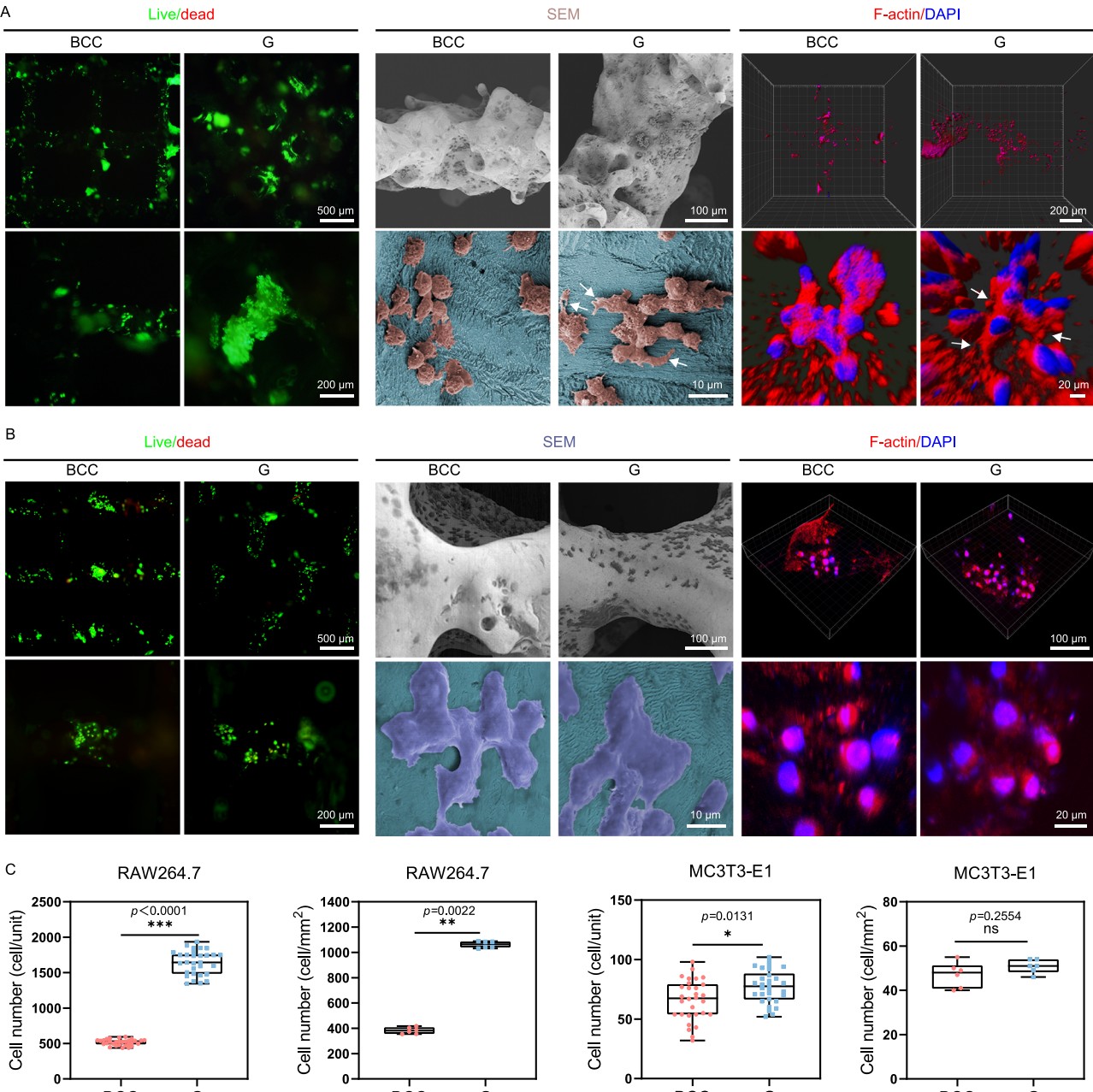

**Fig. 4 | In vitro biocompatibility evaluation for RAW264.7 and MC3T3-E1 cells of Zn-Li porous scaffolds with BCC and G pore units. A** Live/dead staining, SEM images and F-actin staining of RAW264.7 cells on the scaffolds at 48 h. White arrows indicate RAW264.7 cells with extensive filopodia extended. **B** Live/dead staining, SEM images and F-actin staining of MC3T3-E1 cells on the scaffolds at 48 h. **C** Quantitative analysis of number of viable cells per unit from live/dead staining images ($n = 6$, samples per group). Quantitative analysis of number of viable cells per square millimeter from live/dead staining images ($n = 30$, samples per group). For box-whisker plots, box edges correspond to 25th and 75th percentiles, lines inside the box correspond to 50th percentiles, and whiskers include minimum and maximum of all data points. P-values are calculated using one-way ANOVA with Tukey's post hoc test, *$p < 0.05$, **$p < 0.01$, ***$p < 0.005$. Each image was acquired independently three times, with similar results (**A**, **B**). Source data are provided as a Source Data file.

in the acute inflammation phase at 3 days. At 1 month, the inflammatory response at the porous scaffold-tissue interface has largely resolved. Fewer iNOS+ M1 macrophages are detected in the G sample, while an increased presence of CD163+ M2 macrophages is observed at the implant-tissue interface in the G scaffold group compared to the BCC group, both at 3 days and 1 month. More importantly, the ratio of M2/M1 macrophages is consistently higher in G scaffold than in BCC scaffold.

The expression of ALP and OCN evaluate the early repair ability of the bone regeneration process around the implant. Figure 7B shows the localization of ALP in the implant-bone interface, where it is mainly

expressed in osteoblasts in the bone marrow cavity around new bone. OCN is mainly expressed in mature osteoblasts around new bone. The G structure induced a significantly higher ALP positive area than BCC. Further, G structure induced more OCN expression on the surface of new bone than BCC. These histological analyses confirm that osteoblast-like cells are performing bone repair in the defect, with new bone gradually filling the defect and remodeling of the more mature bone matrix over time. The G scaffold facilitates a swift transition of the inflammatory response from M1-type pro-inflammatory to M2-type anti-inflammatory, promoting bone regeneration.

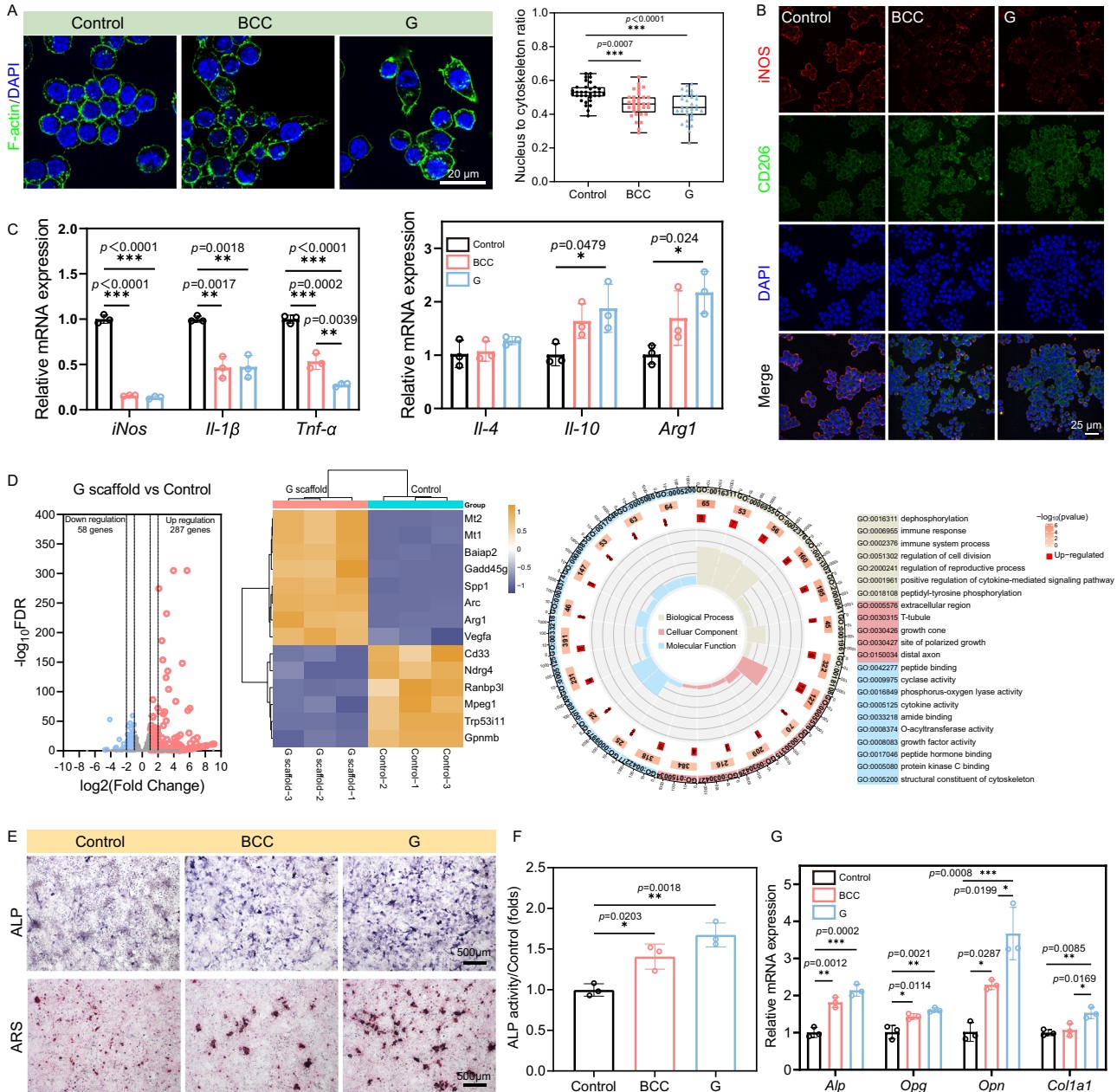

**Fig. 5 | Zn-Li porous scaffolds regulate macrophage polarization to induce osteogenic differentiation in vitro. A** Cytoskeleton staining images of RAW264.7 cultured in BCC and G scaffold extracts for 48 h. Nucleus to cytoskeleton ratio from cytoskeleton staining images (*n* = 33, samples per group). For box-whisker plots, box edges correspond to 25th and 75th percentiles, lines inside the box correspond to 50th percentiles, and whiskers include minimum and maximum of all data points. **B** Representative image of iNOS (M1-type macrophage marker) and CD206 (M2-type macrophage marker) immunofluorescence staining of RAW264.7 cultured in BCC and G scaffold extracts for 48 h. **C** qRT-PCR results of M1-type macrophage cytokines (*iNos*, *Il-1β*, *Tnf-α*) and M2-type macrophage cytokines (*Il-4*, *Il-10*, *Arg1*) (*n* = 3, independent experiments). Data are presented as mean ± standard deviation. **D** RNA sequencing analysis of volcano plot showing the differentially expressed genes (up-regulated genes: red; down-regulated genes: blue) in 5-fold′ diluted G scaffold extracts treated group versus the Control group. A heat map

showing the 14 regulated genes. Go analysis of differentially expressed genes (*n* = 3, samples per group). *P*-values were determined by two-sided *t*-test without adjustments. Significant differential expression was determined using thresholds of padj ≤ 0.05 and |log2 (Fold Change)| ≥ 1.0. **E** ALP staining and ARS staining of MC3T3-E1 cultured in CM supplemented with osteogenic components at 7 and 14 days. ALP: alkaline phosphatase, ARS: Alizarin Red S. **F** ALP activity quantitative of the MC3T3-E1 at 7 days (*n* = 3, independent experiments). Data are presented as mean ± standard deviation. **G** Osteogenesis-related genes mRNA expression levels of MC3T3-E1 detected by qRT-PCR (*Alp*, *Opg*, *Opn*, and *Col1a1*) at 7 days (*n* = 3, independent experiments). Opg Osteoprotegerin, Opn Osteopontin, Col1a1 type I collagen. Data are presented as mean ± standard deviation. P-values are calculated using one-way ANOVA with Tukey's post hoc test, *p < 0.05, **p < 0.01, ***p < 0.005. Each image was acquired independently three times, with similar results (**A**, **B**, **E**). Source data are provided as a Source Data file.

## In vivo bone regeneration

Micro-CT, methylene blue acid fuchsin staining, and second-harmonic generation (SHG) are combining together to evaluate the hard tissue regeneration and ingrowth in the Zn-Li porous scaffolds at 3 months. Qualitatively, scaffolds with the G pore unit are filled with newly

formed bone tissues while less hard tissue can be found in the BCC scaffold (Fig. 8A). Tissue staining confirms new bone ingrowth into the pore area of G scaffold with interconnected features, but few bone tissues are visible in the pore area of BCC scaffold. On the contrary, more degradation products occupy the pore area of the BCC scaffold.

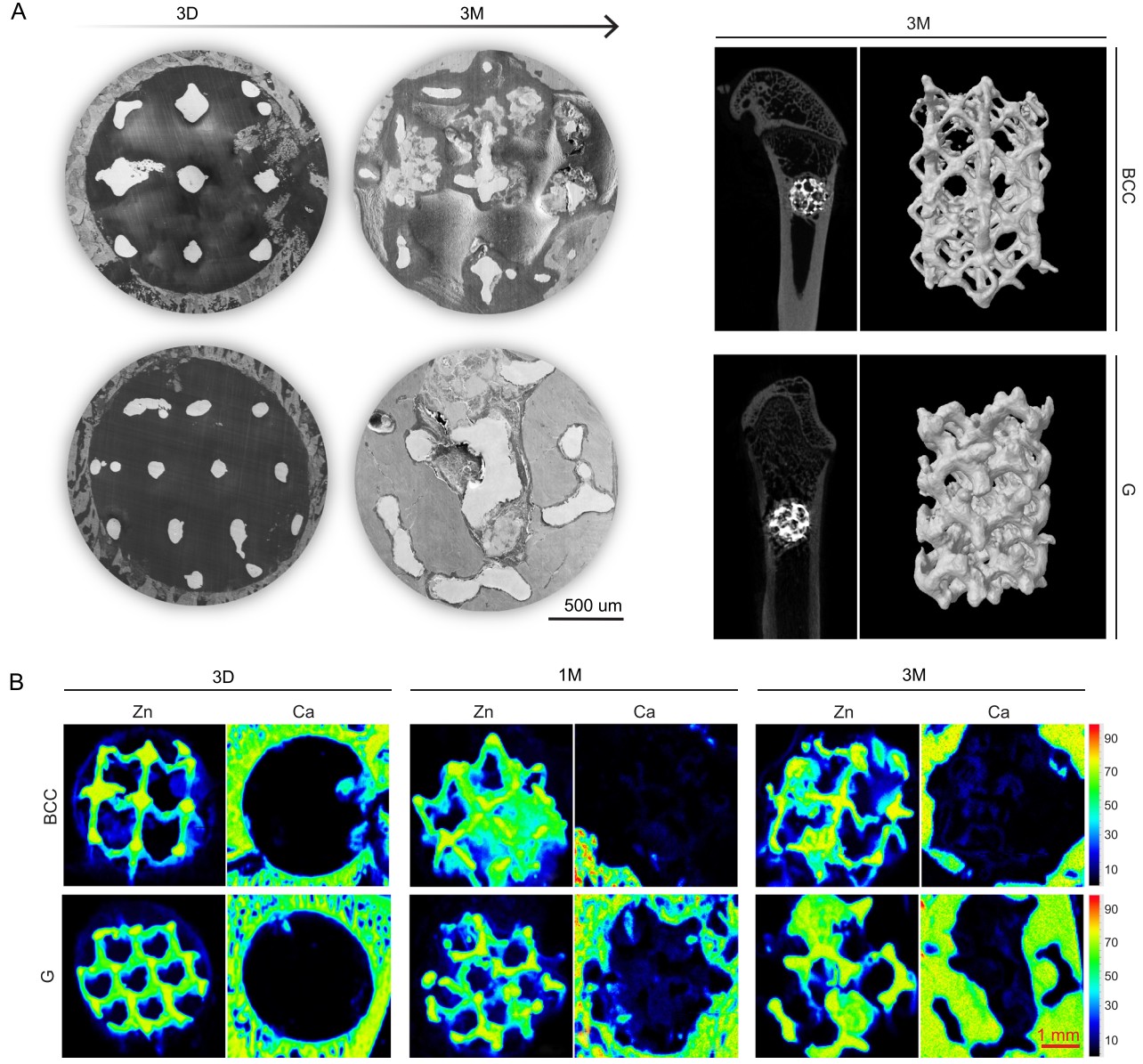

**Fig. 6 | In vivo degradation behavior of Zn-Li porous scaffolds with BCC and G pore unit in femur of rats. A** Cross-sectional images (SEM) of Zn-Li porous scaffolds at 3 days and 3 months. Micro-CT 2D cross sections and 3D reconstruction of the metallic parts of Zn-Li porous scaffolds at 3 months. **B** XRF images of representative cross sections of Zn-Li porous scaffolds. The intensity and distribution of Zn and Ca signals are marked by color ribbon. For Zn, red to yellow indicates surface metallic struts; green indicates degradation products or underlying metallic struts; navy blue indicates the Zn products dispersed in tissues. For Ca, red to green indicates bone tissue; navy blue indicates the mineralized matrix or Ca-P complexes. Each image was acquired independently at least three times, with similar results (**A**, **B**).

Quantitatively, BV/TV in the G scaffold is higher than that of the BCC scaffold significantly while values of Tb. Sp follow the opposite trend (Fig. 8B). Analysis of the staining images supports the results of CT data as well. It is worth noting that the pore size distribution that is conducive to bone ingrowth is in the range of 200-600 μm for all Zn-Li porous scaffolds. The SHG imaging can specifically target type I collagen without destruction, which is performed here to analyze the collagen fiber orientation within the Zn-Li porous scaffolds (Fig. 8C). In general, two objects are visible under the SHG imaging. One is the type I collagen distributed in the tissue, and the other one is zinc oxide located in the scaffold struts. At 1 month, signal is visible only from zinc oxide, and no collagen is found adjacent to the struts. At 3 months, the collagen fibers are well aligned and oriented along the struts in the G scaffold, exhibiting the similar spatial orientation to the scaffold strut geometry. For example, the collagen fiber bundles display a curved feature along the minimal surface structure (G scaffold). In comparison, the type I collagen does not show a clear orientation in the BCC scaffold. In terms of collagen distribution characteristics, point-like distribution in the BCC scaffold replaced bundle-like distribution in the G scaffold. Here, results demonstrate that pore units (pore size, geometry) have a non-negligible influence on the orientation, regeneration, and ingrowth of hard bone tissues.

## Discussion

As a temporary hard tissue substitute material, a bone scaffold must support the defect region for a period of time until the hard callus formation and replacement. For porous bone scaffolds, the better the mechanical properties, the more room is left for regulation of other crucial properties, such as degradation and biological function, when designing the architecture. Here, the Zn-Li alloy system provides a

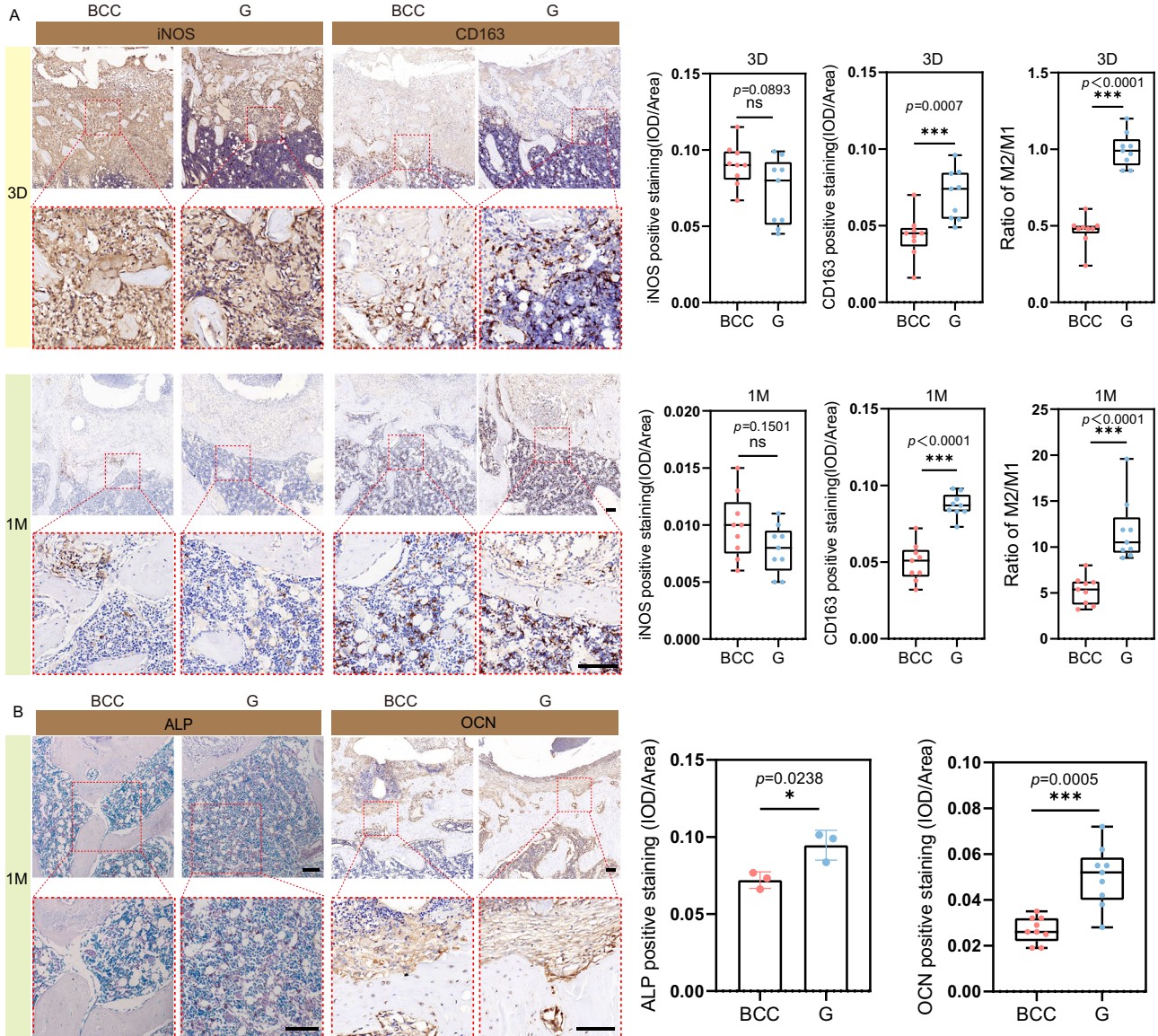

**Fig. 7 | Inflammatory response and early osteogenesis of Zn-Li porous scaffolds in rat femur at 3 days and 1 month. A** Representative image of iNOS (M1-type macrophage marker) and CD163 (M2-type macrophage marker) immunohistochemistry staining of rat femurs histological sections at 3 days and 1 month ($n = 9$, samples per group). The scale bar = 100 μm. Quantitative analysis of iNOS and CD163 implant-tissue interface calculated from immunohistochemistry staining images. **B** Representative images of ALP (osteoblastic lineage cell marker) staining and OCN (mature osteoblast marker) staining images in rat femurs histological sections at 1 month (ALP: $n = 3$ samples per group; OCN: $n = 9$ samples per group). The scale bar = 100 μm. ALP and OCN positive staining calculate from images by using image pro plus 6.0 software. Data are presented as mean ± standard deviation. For box-whisker plots, box edges correspond to 25th and 75th percentiles, lines inside the box correspond to 50th percentiles, and whiskers include minimum and maximum of all data points. *P*-values are calculated using one-way ANOVA unpaired *t*-test with a Mann–Whitney test, *$p < 0.05$, **$p < 0.01$, ***$p < 0.005$. Each image was acquired independently three times, with similar results. Source data are provided as a Source Data file.

platform in terms of mechanical performance based on our previous study[2]. We proved the feasibility and efficacy of using Zn-Li alloy-based bone plates and screws to treat bone fracture in a rabbit femur model even compared with Ti6Al4V counterparts, which is challenging for most of the biodegradable materials[4]. With 0.8 wt.% Li addition, a uniform and ultra-fine microstructure appears which is characterized by alternately arranged $LiZn_4$ phase and Zn phase with interlamellar spacing at about 200-300 nm. As a result, Zn-0.8Li alloy achieves the optimal combination between strength and plasticity. Lithium can significantly modify the degradation and ion release behavior of pure zinc. With more negative potential, $LiZn_4$ phase corrodes preferentially when contacting body fluid followed by forming lithium carbonate ($Li_2CO_3$) (Fig. 1D). Lithium carbonate tends to dissolve in neutral physiological environment and create alkaline microenvironment via

hydrolysis reaction (Fig. 1E). This will immobilize free Zn ions by forming solid products like zinc oxide or zinc carbonates[24]. As a result, the release of Zn ions will be reduced and partially replaced by Li ions. The concentration of Zn decreases by 32% from the Zn extract to the Zn-0.8Li extract, with the Zn:Li ratio shifting to 4:1. To take advantage of this phenomenon, we evaluate the impact of Zn and Li co-release on macrophage polarization systematically before using it as an important way for osteoimmunomodulation. As an essential trace element, Zn is involved in many metabolic reactions[25]. In particular, Zn has a regulatory effect on the bone immune system[26,27]. Previous studies have shown that Zn ions modulate bone immune responses in vitro in a dose-dependent manner. Notably, low dose concentrations (11.25–90 μM) of Zn can promote M0 macrophage polarization towards the anti-inflammatory phenotype macrophage (M2) and

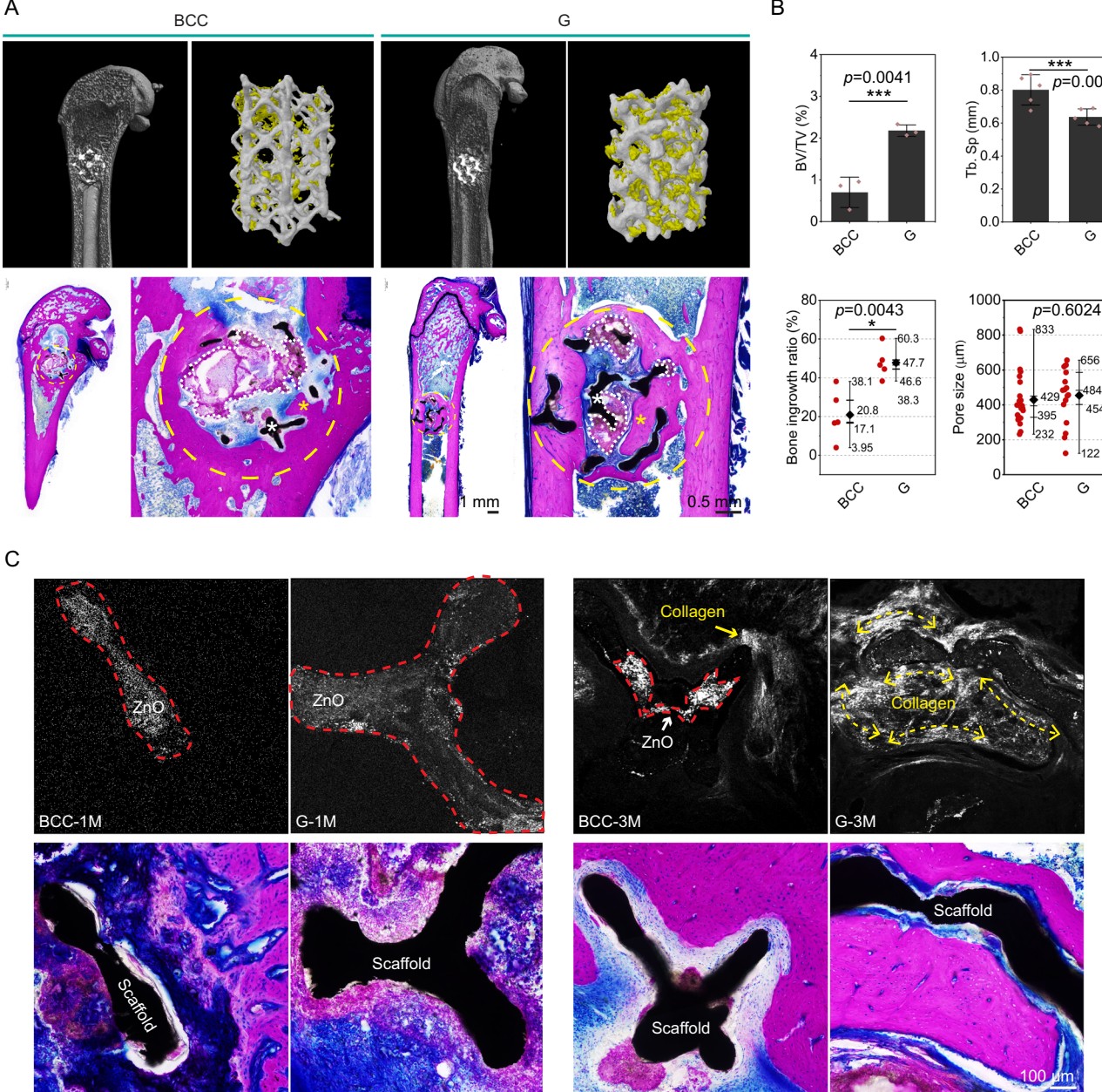

**Fig. 8 | Bone regeneration in Zn-Li porous scaffolds with BCC and G units in femur of rats at 3 months. A** Micro-CT 2D image, and 3D reconstruction of metallic samples (silver gray) and newly formed bone (yellow). Methylene blue acid fuchsin staining of rat femurs and magnifications of implantation sites. Bone tissue is stained in dark purple. White asterisk refers to the scaffold, and yellow asterisk refers to new bone tissue. Yellow dashed circle indicates the area for implantation, and white dashed line area refers to the degradation products. **B** Bone volume/ Tissue volume (BV/TV, $n = 3$, samples per group) and trabecular separation (Tb. Sp, $n = 5$, samples per group) calculated by CT data. Data are presented as mean ± standard deviation. For box-whisker plots, box edges correspond to 25th and 75th percentiles, lines inside the box correspond to 50th percentiles, and whiskers include minimum and maximum of all data points. Bone ingrowth ratio ($n = 5$, samples per group) and bone ingrowth pore size ($n = 20$, samples per group) are measured with 2D images. **C** Second-harmonic generation (SHG) images of scaffolds with corresponding Methylene blue acid fuchsin staining at 1 and 3 months. The red dashed areas indicate the signal of zinc oxide. The collagen fiber orientation is marked with yellow double arrows. Data are presented as mean ± standard deviation. *P*-values are calculated using one-way ANOVA with Tukey's post hoc test, *$p < 0.05$, **$p < 0.01$, ***$p < 0.005$. Each image was acquired independently three times, with similar results. Source data are provided as a Source Data file.

inhibit polarization towards pro-inflammatory phenotype macrophage (M1)[28]. Meanwhile, Li demonstrates bone immunomodulatory prowess by enhancing anti-inflammatory factors like *Arg1*, *Il-4*, and *Il-10*. This prompts a regulation of macrophage polarization towards the M2 and the release of osteogenic factors, thereby promoting osteogenic differentiation in BMSCs[29,30]. Therefore, there should be an optimal combination ratio of Zn and Li ions to modulate bone immunity in Zn-Li alloy system. As shown in Fig. 1F, G, the extract of pure Zn only downregulates the expression of M1-related genes such as *Tnf*-α, *iNos*,

and *Il-1β*. For Zn-Li alloy extracts, the expressions of M2-related genes are upregulated significantly in addition to the downregulation of M1-related genes. Among them, Zn-0.8Li group exhibits the optimal modulation efficiency.

A cutting-edge 3D printing technology is adopted here to fabricate the Zn-0.8Li alloy porous scaffold based on our previous study[11]. The next critical issue that needs to be addressed is the surface quality of the printed scaffold, which is a common issue of 3D printed metal biomaterials. Usually, unmelted powders and defects will cover the

surface of porous scaffolds and play negative role on their biological performance. Few studies have looked into the surface quality of 3D-printed biodegradable metals as coarse surface morphology with large amounts of unmelted powders are commonly seen in 3D-printed Zn-based bone scaffolds, which prevent the new bone tissue from ingrowth and osseointegration[10,12]. From our perspective, surface treatment can be an important way to regulate the early inflammatory immune response of bone injury. Additionally, the alternately arranged $LiZn_4$ and Zn phases with interlamellar spacing at about 200-300 nm lays a good microstructure foundation for creating surface microtopography. Therefore, physical, chemical, or electrochemical means are conducted to create different surface morphology and patterns. With only ultrasonic treatment, large amounts of unmelted powders are still visible on the scaffold surface. After acid etching, most of the loosely attached powders are removed with only half-melted powders left. Meanwhile, a clear lamellar pattern in nanoscale appears on the scaffold surface. Further with EC polishing, scaffold surface is smooth with almost no powder left. Additionally, protruded wavy-like micropatterns with a roughness (Ra) at around 114 nm are predominant on the surface. In early attachment test, RAW264.7 only adheres on the EC polished surface, accompanied by protruded filopodia and a great increase in cell spreading area compared to the other groups (Fig. 2C). Immune cells swiftly arrive at the implant site right after the implantation. Among them, macrophages, pivotal in the foreign body response, quickly secrete pro-inflammatory factors to kickstart a cascading immune response[15,31]. Hence, the pancake-like macrophages with high spreading area and filopodia formation on the EC-polished surface suggests a swift functionalization of immune cells, aided by the nanoscale roughness of the surface pattern. This may offer a favorable early immune microenvironment that promotes tissue regeneration at later stage. A recent study reported a honeycomb-like $TiO_2$ microstructure with 90 nm roughness activated the anti-inflammatory macrophage phenotype, and induced high expression of *Il-4*, *Il-10* anti-inflammatory factors for optimal osseointegration[21].

For biodegradable bone scaffold, the pore geometry will definitely have great impacts on the spatiotemporal characteristics of degradation, which shapes the fate of bone repair and regeneration. Two representative pore units with distinct geometries are selected to illustrate the impact. One is a typical minimal surface geometry-gyroid (G) with biomimetic features that its average mean curvature is similar to trabecular bone[32]. A traditional body centered cubic (BCC) lattice pore unit is used as the control. Key parameters include porosity, surface area, specific surface area, and strut thickness in both units are similar (Table S2). The diffusion test finds that the BCC unit displays a strong anisotropy in terms of diffusion coefficient compared to the G unit. In rat femur, both scaffolds show visible degradation 3 days after implantation as the signal of Zn element spreads over the defect region (Fig. 6B). At 1 month, the degradation of G scaffold remains uniform with most of the pore region available for tissue ingrowth. In contrast, nearly half of the pore region is occupied by solid degradation products in a non-uniform manner in the BCC scaffold. The dynamic immersion test shows that the weight loss of the BCC scaffold is 1.6-fold higher than that of the G scaffold after 28 days in SBF solution. Meanwhile, the degradation of G unit is much more uniform than the BBC unit as shown in the CT image (Fig. 3C). The anisotropic feature and minimal surface geometry can contribute to the proper degradation rate and uniform degradation behavior of G scaffold as proved in electrochemical and ion diffusion studies (Figs. 2B, 6B, and S3). As a result, degradation products, especially the ion release and distribution may orchestrate the inflammatory reactions, which mediates the subsequent bone regeneration. At 3 days post-injury, M1-type macrophages are found predominantly in the BCC scaffold while the M2/M1 ratio of G scaffold is around 1:1 (Fig. 7A). At 1 month, the M2/M1 ratio in G scaffold is more than 2-fold of the BBC scaffold. Meanwhile, early osteogenic factor expression (ALP and OCN) is found significantly

higher in the G scaffold (Fig. 7B). Consequently, the bone regeneration and ingrowth in the G scaffold are significantly better than that of the BBC scaffold at 3 months. Additionally, the curved geometry of G unit serves as a good guidance of the secretion of type I collagen. Large amounts of collagen fibers are well aligned and oriented along the struts in the G scaffold, which is invisible in the BCC unit (Fig. 8). To further understand the mechanism of immunomodulatory osteogenesis, we investigate the impact of Zn-0.8Li scaffold on macrophage polarization and subsequent immunomodulation on osteogenic differentiation and mineralization of pre-osteoblasts (Fig. 5). The extract of the G scaffold effectively promotes RAW264.7 polarization towards the M2 type and the release of anti-inflammatory factors (*Il-4*, *Il-10*, and *Arg1*), while inhibiting the polarization towards the M1 type and pro-inflammatory factors expression (*Tnf-α*) compared to the BCC scaffold. In transcriptomic analysis, the G scaffold activates the associated JAK/STAT pathway in RAW264.7 while down regulates the MAPK pathways. As previously reported, The JAK/STAT pathway is considered to an important signaling pathway for macrophage polarization to the M2 phenotype[33–35]. In comparison, MAPK pathways are the key to activate M1-type macrophage[35,36]. *Il-10* and *Il-4* are crucial cytokine that modulates macrophages to polarize toward M2 phenotype. The binding of *Il-10* to its receptor could activate JAK1 and STAT3[37]. Meanwhile, the binding of *Il-4* to its receptor could activate JAK1 and JAK3. Furthermore, activation of the JAK/STAT signaling in macrophages shifts itself from the pro-inflammatory to the anti-inflammatory phenotype to promote tissue regeneration[38]. Subsequently, the released polarization cytokines promote osteogenic differentiation of MC3T3-E1 cells and accelerate extracellular matrix calcium junction mineralization via up regulation of gene expression in *Alp*, *Opg*, *Opn*, and *Col1a1*.

The Zn-Li bone scaffold studied here offers a significant advantage due to its combination of high strength, biodegradability, bioactivity, and printability. Consequently, its potential clinical applications are targeted toward bone defects at load-bearing sites, such as critical segmental defects in long bones. Current treatments for such bone defects typically involve autologous cortical bone grafts or metallic implants, particularly 3D printed titanium scaffolds[39]. While autologous bone grafting is considered the gold standard, it requires an additional surgical procedure and is often associated with donor site morbidity or insufficient graft material. Titanium exhibits excellent mechanical performance and biocompatibility, facilitating good osseointegration with new bone tissue. Utilizing 3D printing technology, titanium scaffolds can be customized to meet patients' individual needs for bone implants. However, titanium scaffolds remain in place permanently after new bone regeneration and ingrowth, leading to long-term interference with bone remodeling and potential chronic inflammation due to trace Ti ion release. Moreover, titanium scaffolds are not ideal for treating bone defects in complex situations such as osteoporosis or bone infections due to their bioinert nature.

Zn-Li alloys demonstrate superior comprehensive mechanical properties compared to medical-grade pure Ti (Table S4). In a rabbit tibial fracture model, the Zn-Li alloy screw and plate system exhibited comparable performance to Ti-6Al-4V alloy counterparts[4]. Furthermore, it is possible to modulate bioactivities through alloying with functional elements. For instance, alloying with trace amounts of Sr resulted in a Zn-0.8Li-0.1Sr intramedullary nail showing superior osteogenesis-inducing and osteoporotic-bone-fracture-treating effects compared to pure Ti[40]. Zn-Li-Ag demonstrated potent bactericidal effects against methicillin-resistant Staphylococcus aureus (MRSA), leading to remarkable infection control and favorable bone retention in an MRSA-induced rat femoral osteomyelitis model[6]. To compare the osteogenesis performance between 3D printed Zn-Li scaffolds and Ti scaffolds, a critical bone defect was created in rabbit femurs. At 2 months, the newly formed cortical bone volume was significantly higher in the Zn-Li scaffold than in the Ti scaffold, as

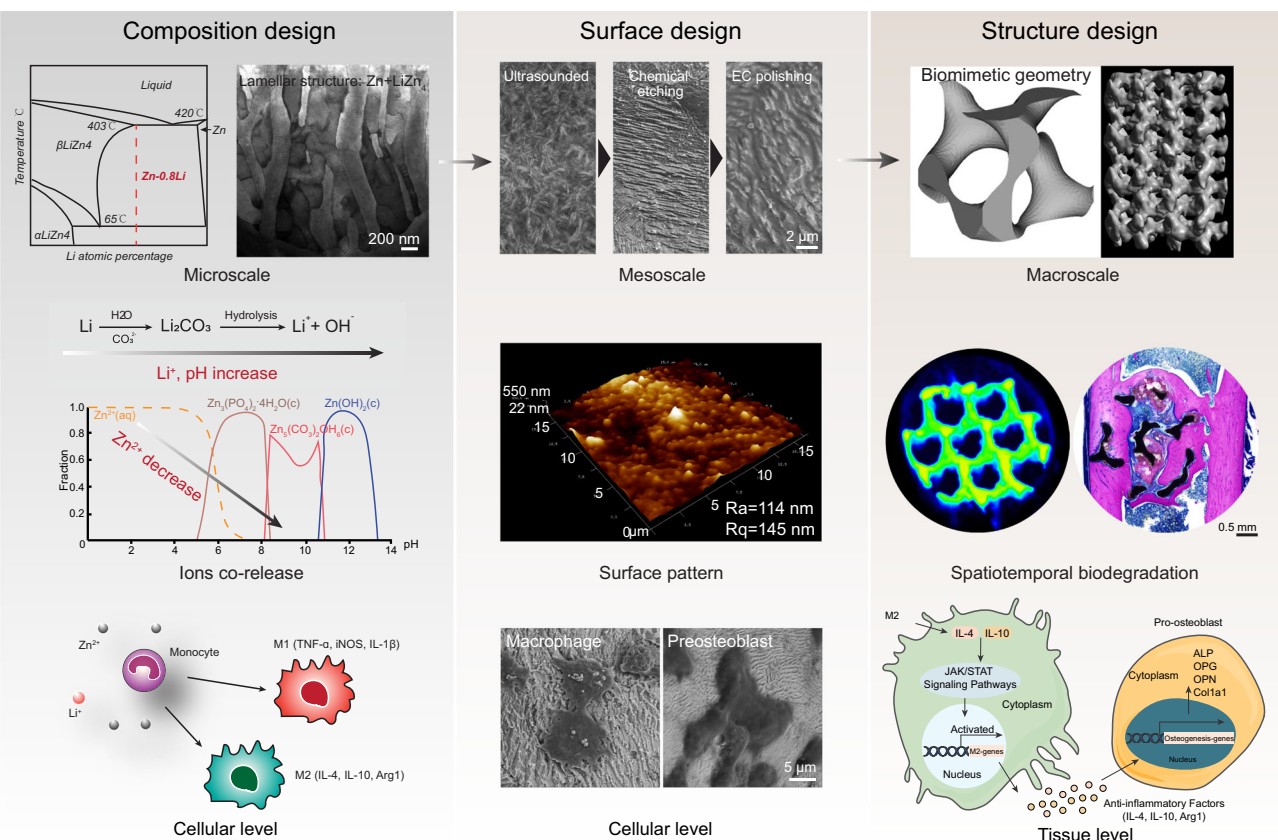

**Fig. 9 | Multiscale integrated design of 3D printed biodegradable Zn-based porous scaffolds for immunomodulatory osteogenesis.** At microscale, 0.8Li is alloying with Zn to form a lamellar microstructure via eutectic reaction, which impacts the corrosive microenvironment by partially replacing Zn ion release with Li ion release, and modulates the polarization of macrophages. At mesoscale, the wavy-like surface pattern with nanoscale roughness initiates the functionalization of macrophages with filopodia formation. At macroscale, the G scaffold with a biomimetic minimal surface geometry shows a spatiotemporal biodegradation behavior, leading to a heightened efficiency in promoting macrophage polarization towards an anti-inflammatory phenotype within 1 month, subsequently leading to significantly collagen deposition and enhanced new bone formation at 3 months. The G scaffold may activate the JAK/STAT pathway in macrophages via up regulating the expression of IL-4, IL-10, which subsequently promotes the osteogenesis.

evidenced by Micro-CT and histology. Compared to pure Ti scaffold, new bone tissue almost refilled the defect region in the Zn-Li scaffold while cortical bone on lateral sides of the defect was obviously thickened (Fig. S7).

In summary, we address the conundrum of balancing rapid degradation and avoiding excessive toxicity through the utilization of osteoinmunomodulation via a multiscale architecture design applied to Zn-based porous scaffolds (Fig. 9). The incorporation of 0.8 wt.% Li in the alloy provides a solid mechanical foundation for scaffold printing. The simultaneous release of Zn and Li ions from the Zn-0.8Li alloy significantly enhances the polarization of macrophages, favoring a pro-regenerative phenotype. The unique structure of the Zn-0.8Li alloy, featuring alternately arranged LiZn$_4$ and Zn phases with inter-lamellar spacing of 200–300 nm, facilitates the creation of nanoscale wavy-like micropatterns through EC polishing. This, in turn, activates macrophages during early attachment, promoting a high spreading area and filopodia formation. The G scaffold, distinguished by its anisotropic features and minimal surface geometry, exhibits an appropriate degradation rate and uniform behavior both in vitro and in vivo. As a result, the G scaffold demonstrates enhanced efficiency in promoting macrophage polarization toward an anti-inflammatory phenotype within one month, leading to significantly elevated osteogenic markers, increased collagen deposition, and enhanced new bone formation at three months. Additionally, the G scaffold may activate the JAK/STAT pathway in macrophages by upregulating the expression of *Il-4* and *Il-10*, subsequently promoting osteogenesis.

## Methods

### Compliance with ethical standards
All surgical procedures were conducted by the ARRIVE guidelines and received approval from the Animal Ethics Committee of the National Research Institute for Family Planning (NRIFH 21-2203-18).

### Materials preparation
The Zn-Li alloy systems were generated using both pure Zn (99.995 wt.%) and a Zn-Li master alloy ingot, following the specified alloy composition (Table S1) by the Hunan Rare Earth Metal Material Research Institute. The nominal and actual compositions of the Zn-Li alloys are detailed below. The as-cast alloys underwent a hot extrusion process, transforming them into 10 mm diameter rods at a temperature of 210 °C with an extrusion ratio of 16:1.

The pre-alloyed Zn-0.8Li cylinder rods underwent atomization into powder particles using argon gas (Nanoval, Germany). The mean powder size, denoted as D50, measured 29.7 μm. The chemical compositions of the powders were analyzed through inductively coupled plasma optical emission spectroscopy (ICP-OES, iCAP6300). The Li element content in the powder was determined to be 0.69 ± 0.01 wt.%.

Prior to the adoption of Laser Powder Bed Fusion (L-PBF) technology, the drying of powders involved subjecting them to a 4-hour vacuum oven treatment at 70 °C to eliminate moisture. The manufacturing of scaffolds was carried out using a commercial L-PBF system (BLT, China), featuring a single-mode ytterbium fiber laser (IPG YLR-500, Germany) with a focus spot diameter of 70 μm at a wavelength of

1070 nm. The processing chamber was filled with argon, maintaining a purity exceeding 99.99%. The oxygen content in the processing chamber was consistently maintained below 120 ppm throughout the L-PBF process. A pure zinc plate, 20 mm in thickness, served as the substrate. Prior to processing, the substrate underwent meticulous grinding and cleaning with ethanol. The crucial processing parameters for L-PBF included laser power (P), scanning speed (vs), hatching space (Hs), and layer thickness (Ds). Following the optimization of processing parameters, Hs and Ds for the Zn-0.8Li alloy were set at 70 and 20 μm, respectively. For the Zn-0.8Li alloy, P and vs were specified as 40 W and 800 mm/s. The inner hatching region employed a zig-zag scanning route with a 67° rotation per layer, while outline contouring was employed to enhance the dimensional accuracy of the scaffolds. Biomimic triple periodic minimal surface Gyroid (G) structure and traditional body-centered cubic (BCC) structure were printed in Φ10×2 mm for in vitro cell test and electrochemical test, Φ6×6 mm for mechanical test and dynamic immersion test, Φ3×4 mm for animal implantation, respectively.

## Surface treatment and characterization

As-printed scaffolds were ultrasound in ethanol for 24 h to remove the unmelted powders. For chemical treatment, scaffolds with different sizes were immersed in an 10% acid solution for 10-60 s. For electrochemical polishing, samples were immersed in phosphoric acid ethanol solution with 1A current applied for 60–120 s and water bathed at 55 °C. The surface morphology was recorded by a SEM. The surface roughness was measured by an atomic force microscopy (AFM, Bruker) with a tapping mode at 0.5 Hz scan rate.

## Microstructure characterizations

As-extruded Zn-0.8Li alloys were cut into 1 mm thickness disk followed by grounding and polishing with 0.25 μm diamond slurry. Polished samples were etched with 4% HNO$_3$/alcohol solution for 5-10 s and pictured by a scanning electron microscopy (Hitachi S-4800, Japan). An X-ray diffractometer (XRD, Rigaku DMAX 2400, Japan) was used to examine the intermetallic phases with scanning range from 10° to 90° at a scan rate of 2° min$^{-1}$ and step of 0.02°. Polished samples were further processed into 60 μm thickness, followed by punching into 3 mm diameter disks, and ion-beam milling using Gatan PIPS 691 with 10 KeV at −25 °C to −30 °C. Samples were plasma cleaned by Gatan SOLARUS 950 before visualized under a high-resolution high-angle annular dark-field mode (HAADF) at 300 kV using a scanning transmission electron microscopy (FEI Titan G2 60–300 ChemiSTEM).

## Mechanical tests

Specimens intended for tensile testing were precision-machined from the Zn-Li extruded rods in accordance with ASTM-E08-04a standards. These specimens were subjected to examination using a universal material testing machine (Instron 5969, USA) operating at a displacement rate of 1×10$^{-4}$ s$^{-1}$. For compressive test, Zn-0.8Li alloy scaffolds with G and BCC units were tested at a displacement rate of $3 \times 10^{-3}$ s$^{-1}$. The determination of yield strength was based on identifying the stress at which 0.2% plastic deformation occurred. The deformed morphologies of samples were visualized using SEM.

## Degradation characterizations

Conducting electrochemical tests involved the utilization of a three-electrode cell comprising a platinum counter electrode, a saturated calomel electrode, and a work electrode. These tests were executed within an electrochemical working station (Autolab, Metrohm, Switzerland). The test chamber was filled with HANKS' simulated body solution (pH 7.4) and maintained at 37 °C via water bath. The sample was immersed for 23 h before recording the open-circuit potential (OCP) (1 h). Electrochemical Impedance Spectroscopy (EIS) measurements were conducted by applying 10 mV perturbation to the OCP

value within a frequency range spanning from 10$^5$ to 10$^{-2}$ Hz. The surface morphology of corroded samples was pictured with a SEM after drying in air. Scanning Vibrating Electrode Technique (SVET) measurements were performed to measure the potential and pH distribution on sample surface in HANKS' solution for 24 h with a 100 μm step size, using the SVET system (Applicable Electronics Inc., USA). The chemical composition of corrosion products was analyzed by an XPS (NAP-XPS, Specs, Germany).

The dynamic immersion platform comprises a test chamber, a peristaltic pump, a water bath, and is interconnected by silicone tubes. HANKS' solution (500 mL) circulated at a perfusion rate of 10 rpm corresponding to the plasma perfusion rates through bone marrow[41]. The system operated continuously for 28 days, with solution renewal occurring every two days. The corroded morphology of the scaffold was visualized using scanning electron microscopy (SEM) and further examined with a Micro-CT scanner (Zeiss Xradia 520 Versa, Germany, 120 kV, 66.7 μA). The 3D reconstruction image was processed using CTvox 3.0 software (Bruker, Germany). The weight loss of scaffold samples was measured after removing corrosion products in a 0.1 M HCl solution. Five scaffold samples were tested simultaneously for comparative analysis.

## Diffusion coefficient measurement

The schematic diagram of the diffusion device is shown in Fig. S8. One compartment (donor) contained ZnCl$_2$ solution, while the other compartment (receiver) was added with the same volume of deionized water. The scaffold was positioned at the center between the two compartments. Prior to starting the experiment, the scaffold was immersed in the ZnCl$_2$ solution for 2 h to ensure thorough wetting. During the experiment, rapid stirring was employed to ensure uniformity of the solutions in both compartments, and the temperature is maintained at 37 °C throughout the process. At different time intervals, 0.5 mL samples were taken from each compartment to avoid the exacerbation of diffusion caused by different volumes. The concentration of Zn ions in the samples was detected by ICP.

The zinc ion flux through the scaffold per unit time serves as an indicator of scaffold porosity and interconnectivity. Permeability is a measure of diffusion over time and can be calculated through the *flux* equation[42]:

$$flux(g \cdot m^{-2} \cdot s^{-1}) = \frac{[(C_{acceptor(g \cdot m^{-3})} \times V_{acceptor(m^3)})/S(m^2)]}{time(s)} \quad (1)$$

where $C_{acceptor}$, Zinc ion concentration in acceptor compartment, $V_{acceptor}$, the volume of the acceptor compartment and S, the surface area of the scaffolds. The zinc ion permeability can be calculated:

$$permeability(m^2 \cdot s^{-1}) = \frac{flux(g \cdot m^{-2} \cdot s^{-1})}{\Delta C(g \cdot m^{-3})} \times l(m) \quad (2)$$

$_{where}$ $\Delta C$ is the concentration difference between the donor and the acceptor and $l$ is the average length of the scaffolds. Further, the diffusion coefficient is determined by the obtained permeability according to the following formula:

$$diffusion\ coefficient(m^2 \cdot s^{-1}) = permeability(m^2 \cdot s^{-1}) \times K \quad (3)$$

with $K$ being the partitioning coefficient. Because the Zinc ion radius is far less than in comparison to the pore size, $K$ can be assumed to be equal to one.

## Cytocompatibility

Mouse-derived preosteoblasts (MC3T3-E1, CL-0378) and macrophage cells (RAW264.7, CL-0190) were procured from Wuhan Prucell Life Sciences Co. (Wuhan, China). The MC3T3-E1 cells were cultured in

MEM-α (Gibco) supplemented with 10% fetal bovine serum (FBS) and 1% antibiotic-antimycotic (100 U/mL penicillin and 0.1 mg/mL streptomycin) at 37 °C in a humidified atmosphere with 5% CO2. The RAW264.7 cells were cultured in DMEM (Gibco) supplemented with 10% FBS and 1% antibiotic-antimycotic (100 U/mL penicillin and 0.1 mg/mL streptomycin) under similar conditions. The culture medium was refreshed every two days. MC3T3-E1 cells were passaged using 0.25% trypsin with EDTA upon reaching 90% confluence, while RAW264.7 cells were passaged by gentle scraping of the culture dish.

### Direct contact in vitro biological studies

Zn-Li scaffolds (Φ10×2 mm) were meticulously seeded with 100 μL of RAW264.7 or MC3T3-E1 cell suspensions containing $1 \times 10^6$ cells per sample for 2 h to ensure proper cell attachment. The Zn-Li scaffolds were transferred into a 24-well plate, and 2 mL of fresh culture medium was added to each well.

A live/dead staining kit (PF00008, proteintech) was employed to assess the proliferation of MC3T3-E1 and RAW264.7 cells on Zn-Li scaffold samples (Φ10×2 mm). The cells were cultured on the samples for 48 h and washed with PBS twice. Live/dead staining solution was prepared by mixing 2 mM Ethidium homodimer-1 (EthD-I) and 4 mM Calcein AM with PBS. Subsequently, the porous scaffolds were immersed directly into the staining solution and incubated at room temperature for 20 min to allow for complete coloration. The cells were then washed twice with PBS. The fluorescent images were acquired by a fluorescence microscope (Nikon, Japan) at various magnifications.

The Image Pro Plus 6.0 software (Media Cybernetics) was employed to quantify the number of live cells in images obtained from live/dead staining. Specifically, the number of live cells per scaffold unit was counted in 33 randomly chosen areas, and the number of live cells per square area was quantified in 6 randomly selected areas.

Cytoskeletal staining was utilized to assess the morphology of RAW264.7 and MC3T3-E1 cells on the scaffolds (Φ10×2 mm). The cells were cultured on the samples for 48 h, followed by washing the samples three times with PBS and fixation in 4% paraformaldehyde solution for 20 min at room temperature. Subsequently, the cells were permeabilized with 0.2% Triton X-100 for 10 min. Afterwards, DAPI (C1005, Beyotime) and 200 nM TRITC-labeled Phalloidin (CA1610, Solarbio) were used to stain the nuclei and cellular F-actin for 10 min and 30 min, respectively. Cells were washed by PBS to remove the residue dye. Finally, the cellular morphology on the scaffolds was examined using confocal microscopy (Dragonfly 200).

SEM images were utilized to examine the adhesion of RAW264.7 and MC3T3-E1 cells on the scaffolds (Φ10×2 mm). Following incubation for 48 h, the cells were fixed in a 2.5% glutaraldehyde solution for 2 h at 4 °C. Subsequently, the samples were dehydrated using a gradient of diluted alcohol (50%, 60%, 70%, 80%, 90%, 95%, and 100%) and then left to dehydrate overnight in a 24-well plate. Finally, the cellular morphology was assessed using scanning electron microscopy (SEM).

RAW264.7 were cultured on G scaffold treated in different ways of surface treatment for 6 hours. Cells were fixed in 2.5% glutaraldehyde solutions for 2 h in 4 °C, and dehydrated in gradient diluted alcohol (50%, 60%, 70%, 80%, 90%, 95%, and 100%), then dehydrated overnight in 24-well plate. Cell adhesion and morphology were recorded by SEM and cytoskeletal morphology was determined by F-actin staining. Step of the SEM and F-actin staining experimental procedure were performed as described above.

### Indirect contact in vitro biological studies

The International Organization for Standardization method (ISO10993-12) was employed in this study to prepare extracts of Zn-Li alloys. Briefly, the Zn-Li alloys were sterilized using UV light for half an hour on each side and then incubated in a medium with a volume-to-area ratio of 1.25 mL/cm² at 37 °C for 24 h. Subsequently, the

supernatants were collected and stored at 4 °C. The concentration of Zn ions and Li ions in the alloys was determined using inductively coupled plasma (ICP) analysis.

RAW264.7 cells were seeded at a density of $1 \times 10^4$ cells per well in a 24-well plate. After 24 h of culture, the cells were exposed to a diluted 5-fold extract medium (Fig. S9) and cultured for 48 h. Macrophage polarization was evaluated using immunofluorescence staining and quantitative reverse transcription polymerase chain reaction (qRT-PCR).

Cytoskeletal staining was employed to assess the polarization morphology of RAW264.7 cells. After washing three times with PBS, the cells were fixed with a 4% paraformaldehyde solution for 20 min at room temperature and permeabilized with 0.2% Triton X-100 for 10 min. DAPI (C1005, Beyotime) and 100 nM FITC-labeled Phalloidin (CA1620, Solarbio) were used to stain the nuclei and cellular F-actin for 10 min and 30 min, respectively. RAW264.7 cells were washed by PBS to remove the residue dye. Finally, F-actin (stained green) and nuclei (stained blue) were observed using confocal microscopy (Dragonfly 200). The nucleus to cytoskeletal area ratio was quantified using Image Pro Plus.

Two experimental groups were utilized to elucidate the mechanisms underlying macrophage polarization in response to treatment with 5-fold scaffold extracts: control group (untreated) and the G scaffold group (treated with 5-fold scaffold extracts). RNA extraction from RAW264.7 cells was performed followed by transcriptome sequencing to investigate gene expression profiles. RNA-seq analysis were performed using the free online platform of http://www.bioinformatics.com.cn.

Conditioned medium (CM) was prepared as previously reported[43]. the fresh supernatant from RAW264.7 cells treated with BCC and G scaffold extracts for 48 h was initially collected. The supernatant underwent centrifugation for 5 min to remove residual cells, followed by filtration through a 0.22 μm filter (Millipore, Ireland), and subsequent storage at −80 °C for future use. This supernatant was then combined with fresh MEM-α medium in a 1:2 ratio to produce conditioned medium. In the case of MC3T3-E1 cells, CM medium containing osteogenic components (10 mM β-glycerophosphate and 0.25 mM ascorbic acid) was employed to induce osteogenic differentiation. Fig. 9.

MC3T3-E1 cells were seeded in 24-well plates at a density of $1 \times 10^4$ cells per well. Following 7 and 14 days of osteogenic induction in conditioned medium (CM), alkaline phosphatase (ALP) and alizarin red S (ARS) staining were performed to assess osteogenic differentiation. The CM was refreshed every other day. Briefly, after 7 days of culture, MC3T3-E1 cells were washed twice with PBS, fixed with 4% paraformaldehyde solution for 20 min. The BCIP/NBT ALP color development kit (CA1620, Solarbio) was used to stain ALP activity as described previously[44]. For Alizarin Red S staining, Alizarin red solution (C0148, Beyotime) was used to stain calcium nodules for 10 min to analyze calcification characteristics. Images of the stained cells were captured to evaluate osteogenic properties.

After 7 days of culture in CM, MC3T3-E1 were washed three times with PBS. Subsequently, the cells were lysed using RIPA lysis buffer, and the resultant proteins were collected. The ALP activity was measured according to the ALP quantitative analysis kit protocol (A059-2, Nanjing Jiancheng Bioengineering Institute, China). The total protein concentration was also determined using a BCA protein detection kit to normalize the obtained data. Each group was tested in triplicates.

Quantitative real-time polymerase chain reaction (qRT-PCR) was conducted following a 2-day culture of RAW264.7 cells and a 7-day culture of MC3T3-E1 cells. Total RNA was extracted from the cells using Trizol reagent (Invitrogen), and the quantity and quality of the RNA were assessed using a Nanodrop 2000 Spectrophotometer (Thermo Scientific, USA). Subsequently, complementary DNA (cDNA) templates were synthesized using the PrimeScript RT Reagent Kit

(RR047A, Takara). Following the manufacturer's protocol, real-time qPCR was performed using TB Green® Premix Ex Taq™ II (RR420A, Takara) in a CFX96™ Real-Time System (Bio-Rad Laboratories, Hercules, CA). The $2^{-\Delta\Delta Ct}$ method was employed to analyze the relative mRNA expression levels, with glyceraldehyde-phosphate dehydrogenase (*Gapdh*) as the reference gene. The primer sequences of the genes used in the study are provided in Table S5.

Immunofluorescence staining was conducted after 48 h for RAW264.7 cells and 7 days for MC3T3-E1 cells. The cells were fixed with 4% paraformaldehyde for 30 min, permeabilized with 0.2% Triton X-100 for 10 min, and then blocked with 5% bovine serum albumin (BSA) for 1 h. RAW264.7 cells were incubated with primary antibodies against iNOS (1:200 dilution, Proteintech; 22226-1-AP; Polyclonal) and CD206 (1:200 dilution, Santa Cruz; sc-58986;Monoclonal), while MC3T3-E1 cells were incubated with primary antibodies against RUNX2 (1:200 dilution, Cell signaling technology; #12556; Monoclonal) and OSX (1:200 dilution, Abcam; ab209484; Monoclonal), each at a 1:200 dilution, at 4 °C overnight. Subsequently, the cells were incubated with Fluor 647-conjugated anti-mouse (1:400 dilution, Santa Cruz; sc516244) or Alexa Fluor 488-conjugated anti-rabbit (1:200 dilution, Proteintech; SA00013-2) secondary antibodies for 1 h, followed by incubation with DAPI for 20 min for nuclear staining. The stained cells were visualized using confocal microscopy.

### In vivo study

All surgical procedures were conducted by the ARRIVE guidelines and received approval from the Animal Ethics Committee of the National Research Institute for Family Planning (NRIFH 21-2203-18). A rat femoral condyle defect repair model was utilized to assess the degradation behavior and osteogenesis surrounding the implantation of BCC and G scaffolds (Φ3×4 mm). For the implantation experiment, 12-week-old male Sprague-Dawley rats weighing 350 to 400 g were procured from Beijing Huafukang Biotechnology Co., LTD (SCXK(JING)2019-0008). The surgical procedures were performed under sterile conditions. Briefly, the rats were anesthetized via intraperitoneal injection of ketamine (10 mg kg$^{-1}$) and 2% xylazine (5 mg kg$^{-1}$). Each rat was then positioned with the knee joint in a maximally flexed position, and the right hind limb was shaved and depilated. The femoral condyle was exposed through a combination of blunt and sharp dissection of the muscles, with subsequent removal of the periosteum to expose the surgical site. A cylindrical defect measuring 3×4 mm was surgically created in the femoral condyle of each rat and washed with 0.9% sterile sodium chloride solution. Following this, scaffolds (BCC, G) were implanted, and the wound was closed using surgical sutures. The rats were euthanized at 3 days, 1 month, and 3 months post-implantation. The entire implanted femur was collected and fixed in 4% paraformaldehyde solution for 72 h.

The femur tissue containing the implant was dehydrated for non-decalcified sections using a gradient of ethanol. Subsequently, the specimens were embedded in methyl methacrylate (MMA). The embedded specimens were then sectioned into slices with a thickness of 200 μm and further ground to achieve a thickness of 50 μm.

For decalcified sections, the femur containing the implant was subjected to decalcification in EDTA decalcifying solution (pH 7.2) for 6−8 weeks until reaching an elastic state. During this process, the specimens were regularly replaced with fresh decalcification solution every week. Following decalcification, the specimens underwent serial dehydration and paraffin embedding. Longitudinal sections of the femur tissue, each measuring 5 μm in thickness, were prepared for subsequent histological analysis.

The MMA embedded slices were wire-cut into 2 cm×2 cm sample (1 mm thickness). Samples were grounded and polished before drying in air. Then, sample surface was coated with a layer of gold and recorded using a SEM.

The MMA embedded slices were examined using an X-ray fluorescence (XRF, Bruker M4 Tornado, Germany). The X-ray generator functioned at 50 kV with a current of 600 μA. To minimize background interference, a filter composition of 100 μm of aluminum, 50 μm of titanium, and 25 μm of copper was employed. Elemental mappings were executed in a vacuum environment of 20 mbar directly on the sectioned samples. The lateral step size was set at 20 μm, and each pixel was exposed for 5 ms. For XRF data acquisition and processing, the instrument's proprietary Bruker software was utilized. Quantification of the results was conducted through the fundamental parameter method designed for bulk samples.

The non-decalcified slices were stained with methylene blue acid fuchsin, and visualized under Pannoramic MIDI (DHISTECH). For quantitative analysis, an area containing the scaffold with a diameter of 3 mm was set as the ROI. The Bone ingrowth ratio is defined as the new bone area divides the ROI area ($n = 9$). The pore size of bone ingrowth was measured as the minimum distance between two struts ($n = 20$). Second-harmonic generation (SHG) images was recorded using a confocal FLIM instrumentation (ISS, USA). The SHG signal was generated with a wavelength of 860 nm and detected in the rage of 420-430 nm (half excitation wave length). Images were recorded using a 10× objective.

Paraffin-embedded tissue sections were deparaffinized by placing them in an oven at 67 °C for 2 h, followed by immersion in fresh xylene for 8 min, repeated twice, and subsequently in graded alcohol (100%, 90%, 80%, and 75%) for 5 min each. Finally, the sections were washed with distilled water three times.

The sections were then incubated overnight at 4 °C with primary antibodies against iNOS (1:200 dilution, Proteintech; 22226-1-AP; Polyclonal), CD163 (1:200 dilution, Proteintech; 16646-1-AP; Polyclonal), and OCN (1:200 dilution, Proteintech; 23418-1-AP; Polyclonal), followed by incubation with horseradish peroxidase at room temperature in the dark for 1 h. The sections were imaged using Pannoramic MIDI (DHISTECH). The mean intensity of iNOS, CD163, and OCN staining was quantified as the integrated optical density (IOD) in the area of positive cells, expressed as IOD per area of positive cells (IOD/area), using Image-Pro Plus 6.0 software (Media Cybernetics). At least three random fields were evaluated in each section, and each group consisted of at least three rats.

Alkaline phosphatase (ALP) staining of femur tissue sections was conducted according to the standard protocol (Wako-294-67001). Brown staining in the cells indicated ALP-positive cells, signifying activated osteoblasts. The sections were visualized using a stereo-microscope (Olympus).

Femur with scaffolds were first scanned using a Micro-CT system (Skyscan1176, USA Bruker). Implanted rat femurs were examined with the 18 μm resolution protocol (40 kV, 250 uA). The CT scanning images were reconstructed by using NRecon software (Bruker microCT, Kontich, Belgium) and further analyzed by CTAn, CTVol, and CTVox software to produce 3D images and bone morphometry analysis including new bone formation (BV/TV), bone ingrowth, and trabecular separation (Tb.Sp).

### Statistical analysis

The numerical data were evaluated using GraphPad Prism Version 8.4 software (GraphPad Software, La Jolla CA, USA) and Origin 2019b (Originlab, Northampton MA, USA) by one-way analysis of variance (ANOVA) followed by post hoc Tukey's multiple comparison test. The data were presented as mean ± SD ($n \geq 3$, independent samples) and a difference of $*p < 0.05$, $**p < 0.01$, and $***p < 0.005$ were considered as statistically significant.

### Reporting summary

Further information on research design is available in the Nature Portfolio Reporting Summary linked to this article.

## Data availability

The authors declare that all data supporting the findings of this study are available within this paper and its Supplementary Files. The RNA-seq data generated in this study have been deposited in the NCBI Gene Expression Ominubus (GEO) database under accession number GSE262010 (URL). Source data are provided with this paper.

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

## Acknowledgements

Y.Wu at Boyue Instruments (Shanghai) Co., Ltd. offered great help in analyzing the results of XRF data. We thank Dr. Yan Guan and Dr. Wei Pan for their help with SHG imaging and AFM measurement. Work in the Yang group was supported by grants from Natural Science Foundation of China (Grant No. 52101282) and Fundamental Research Funds for the Central Universities (Grant No. YWF-23-YG-QB-039). Work in the Zheng group was supported by grants from Natural Science Foundation of China (Grant Nos. U22A20121, 5231101024, and 51931001). Work in the Wang group was supported by grants from Natural Science Foundation of China (Grant Nos. 82350003 and 92049201). Work in the Wen group was supported by grants from Natural Science Foundation of China (Grant No. 52175274) and Beijing Hospitals Authority Clinical Medicine Development of Special Funding Support, code: YGLX202337.

## Author contributions

Conceptualization: H.T.Y. and Y.F.Z. Methodology: S.L., H.T.Y., X.H.Q., Y.Q., A.B.L., G.B., H.H., C.Y.S., J.B.D., J.L.T., J.H.S., Y.G., W.P., X.N.G. and B.J. Investigation: S.L., H.T.Y., X.H.Q., Y.Q., A.B.L., C.Y.S., J.B.D. and J.H.S. Visualization: H.T.Y. and S.L. Supervision: H.T.Y., P.W., X.G.W. and Y.F.Z. Writing—original draft: H.T.Y. and S.L. Writing—review & editing: H.T.Y., S.L., P.W., X.G.W. and Y.F.Z.

## Competing interests

The authors declare no competing interests.
