## [Peer Review File · Nature Communications]

Reviewers' Comments:

Reviewer #1:

Remarks to the Author:

Summary: In the manuscript entitled "Multiscale architecture design of 3D printed biodegradable Zn-based porous scaffolds for immunomodulatory osteogenesis", the authors aimed at establishing the combined effects of material composition and pattern surface on the macrophage inflammatory response and osteoblasts' osteogenic differentiation via osteoimmunomodulation. Through the employment of various techniques, the authors have evaluated the mechanical characteristics of the designed structures and observed that by incorporating 0.8 wt.% Li into the Zn alloy the mechanical strength and corrosion time have increased, thus offering a solid foundation for scaffold printing. Moreover, the Zn-0.8Li alloy presents alternatively arranged LiZn₄ and Zn phases which allows for an easier fabrication of nanoscale wavy-like micro-patterns on its surface via EC polishing, characteristic which in turn leads to an improvement in the alloy's degradation rate and biocompatibility. From the in vitro studies conducted with the RAW 264.7 murine like-macrophages and MC3T3-E1 murine pre-osteoblasts, the authors concluded that Zn-0.8Li alloy with a gyroid (G) structure could be the superior alloy in terms of macrophage polarization and the subsequent immunomodulation of the osteogenic differentiation process. In addition, the in vivo studies further confirmed the G structure's ability to elicit an anti-inflammatory response, favourable for an improved tissue regeneration, as proved by the elevated osteogenic markers (increased collagen synthesis and calcium nodules' deposition). Therefore, in their study the authors demonstrate that the employment of a gyroid micro-pattern on the surface of the Zn-0.8Li alloy constitutes a feasible strategy for the development of biomaterials that can optimally balance a superior strength with an immunomodulatory effect.

1) The abstract and conclusions sections do indeed convey the main findings of the study, being clear and concise. However, even if the introduction is appropriately and provides a scientific context for the approached topic, it still lacks important background information regarding the implication of macrophages in the bone tissue regeneration process, the tightly connected relationship between the macrophages and bone cells and the importance of osteoimmunomodulatory properties that biomaterials should possess. This will increase the scientific value of the manuscript and provide a smoother lead-in to the experimental investigations. In that sense the following works should help the authors in gathering vital information: doi:10.1039/D0TB01379J; doi:10.3390/ma14061357; doi: 10.1093/rb/rbaa006.

2) In terms of methodology, even though the authors have employed a broad range of techniques in their investigations, the information regarding the used protocols is insufficient and presents major lacunas, therefore making it harder for other researchers to follow and apply the same methodology in their own studies. With this in mind, for the results to be reproduced and to assure a full transparency, I would recommend that the whole methodology section should be rethought and rewritten, offering supplementary information. For example, the Live&Dead staining, cytoskeleton labelling and osteogenic differentiation investigations (ALP and ARS staining) should be submitted to various modifications where either a step by step protocol or the staining solution names should be offered. If the protocol has been explained in detail in a previously published article, a reference could also be provided in its stead. Moreover, I would recommend that the in vitro biological investigations to be clearly separated into direct contact studies, where the cells have been seeded directly on to the surface of the analysed samples, and indirect contact studies, where extracts and conditioned medium were used. In this way, the methodology will be easier to understand and follow since it will have a logical flow in the experimental steps. A modification in sequence should also be applied to the results section where the in vivo studies should follow the in vitro studies.

3) In the results section, the quality of the presented data is quite good and the obtained results are accurately interpreted, however with some irregularities. For example, for both cell types the cytoskeleton fluorescence images are not very conclusive (Figure 2 and Figure 4), therefore discussion regarding cell shape and actin organization on the aforementioned figures is very hard. I would recommend that the authors should replace the images with appropriate ones or if not possible arrows pointing to said structures, should be added. The same observation for the SEM images, where arrows pointing towards the cells should be added. Moreover, the figures legends

are too simple, and since the article is intended for a broad range of individuals, a more detailed description is required.

In addition, even though the statistical analysis is accurate, the error bars for each graphic are not defined in the corresponding legends, therefore I recommend that the legends should also be modified accordingly in order to include a detailed definition of the error bars for each graphic.

4) Even though the conclusive remarks and the data interpretation are valid and reliable, a lacking in the comparative analysis is observed in the discussion section, therefore a more extensive comparison with the already published literature should be added in order to better highlight the novel aspects and advantages of the proposed Zn-based structures.]

5) For suggested improvements that could help strengthening the work presented in the current manuscript I would recommend the following:

a) A quantitative test such as MTT or CCK-8 in order to assess the cytotoxic effect of both the alloy and its extract since the Live&Dead analysis is a qualitative viability test that offers a limited view on the cellular viability and proliferative processes. In addition, the fluorescence images offered are not very conclusive.

b) A negative control for the macrophage polarization study since from what it can be seen from Figure 8, the cells expressed positive signals for both phenotype markers. Thus in order to eliminate the idea of auto-fluorescence or non-specific labelling, a negative control should be provided for comparison. Another suggestions could include the use of a double labelling or the replacement of a less sensitive marker.

c) In conclusion, the present study is very complex and the concluding remarks are very interesting and quite significant in the context of developing implantable biomaterials with immunomodulatory properties for a proper bone tissue regeneration. It is very clear that the authors belong to a research group with a significant expertise in this domain, therefore the research could be relevant and interesting for various people such as PhD students, young researchers or interdisciplinary teams starting their work in the osteoimmunology field with bone implantable biomaterials for regenerative applications.

Reviewer #2:

Remarks to the Author:

Reviewer #3:

Remarks to the Author:

This work presents an innovative and comprehensive approach to addressing the intricate challenges associated with balancing degradation and toxicity in biodegradable materials, particularly in the context of porous scaffolds for bone tissue engineering. The research approach is logically sound, the data are comprehensive and complete, and the argumentation is solid. Nevertheless, there are improvements that should be made to enhance the quality of the current manuscript.

1. The study effectively introduces the research focus on balancing degradation and toxicity in Zn-based porous scaffolds through osteoimmunomodulation. However, it would be beneficial to provide more context regarding the significance of this conundrum and its relevance to current challenges in tissue engineering.

2. The authors mention that rapid degradation and excessive toxicity pose a dilemma for biodegradable materials when transitioning from bulk to porous forms. However, they do not demonstrate the difference in degradation behavior and biocompatibility between bulk Zn-Li alloy and Zn-Li alloy scaffolds. I suggest providing more data to support this assertion.

3. To achieve a biodegradable scaffold with immunomodulatory capabilities, the authors propose a design strategy that includes chemical composition, surface treatment, and geometry. How do you rank the importance of these factors, and why?

4. In Figure 2, the surface patterns between acid etching and EC polishing are quite similar in terms of morphology and roughness, but results of macrophage attachment are markedly different. Could you please provide further explanation of this phenomenon?
5. In many published studies, zinc exhibits cytotoxicity on osteoblast cells. In Figure 4, MC3T3-E1 cells were well attached to the scaffold struts. What factors do you believe contribute to this difference?
6. The sample size (n) in Figure 3A, 4C, 6, and 8 should be included in the figure legend. Gene names mentioned in this study require thorough checking and should appear in italics.
7. It is intriguing that scaffolds with different geometries (G and BCC) degrade in distinct manners. I suggest the authors discuss this aspect in more detail as it will illuminate how to properly design degradable scaffolds for other researchers.
8. There is a notable disparity in the amount of newly formed bone observed inside the pores of the G scaffold compared to the nearly absent bone tissue found in the BCC scaffold, which presents a stark contrast. What factors contribute to this outcome?
9. It appears that when Zn-based bone scaffolds degrade rapidly, they may interfere with regular bone repair and lead to the failure of bone healing. How can this be prevented from happening from a clinical translation perspective?
10. While the study highlights the enhanced efficiency of the G scaffold in promoting anti-inflammatory macrophage polarization and osteogenesis, it would be beneficial to discuss the potential clinical applications of this technology and outline future research directions aimed at optimizing scaffold design and performance for specific tissue engineering applications.

Response to Reviewer

Reviewer #1 (Remarks to the Author):

In the manuscript entitled “Multiscale architecture design of 3D printed biodegradable Zn-based porous scaffolds for immunomodulatory osteogenesis”, the authors aimed at establishing the combined effects of material composition and pattern surface on the macrophage inflammatory response and osteoblasts osteogenic differentiation via osteoimmunomodulation. Through the employment of various techniques, the authors have evaluated the mechanical characteristics of the designed structures and observed that by incorporating 0.8 wt.% Li into the Zn alloy the mechanical strength and corrosion time have increased, thus offering a solid foundation for scaffold printing. Moreover, the Zn-0.8Li alloy presents alternatively arranged LiZn₄ and Zn phases which allows for an easier fabrication of nanoscale wavy-like micro-patterns on its surface via EC polishing, characteristic which in turn leads to an improvement in the alloy’s degradation rate and biocompatibility. From the in vitro studies conducted with the RAW 264.7 murine like-macrophages and MC3T3-E1 murine pre-osteoblasts, the authors concluded that Zn-0.8Li alloy with a gyroid (G) structure could be the superior alloy in terms of macrophage polarization and the subsequent immunomodulation of the osteogenic differentiation process. In addition, the in vivo studies further confirmed the G structure’s ability to elicit an anti-inflammatory response, favourable for an improved tissue regeneration, as proved by the elevated osteogenic markers (increased collagen synthesis and calcium nodules’ deposition). Therefore, in their study the authors demonstrate that the employment of a gyroid micro-pattern on the surface of the Zn-0.8Li alloy constitutes a feasible strategy for the development of biomaterials that can optimally balance a superior strength with an immunomodulatory effect.

Comment No.1: The abstract and conclusions sections do indeed convey the main findings of the study, being clear and concise. However, even if the introduction is appropriately and provides a scientific context for the approached topic, it still lacks important background information regarding the implication of macrophages in the bone tissue regeneration process, the tightly connected relationship between the macrophages and bone cells and the importance of osteoimmunomodulatory properties

that biomaterials should possess. This will increase the scientific value of the manuscript and provide a smoother lead-in to the experimental investigations. In that sense the following works should help the authors in gathering vital information: doi:10.1039/D0TB01379J; doi:10.3390/ma14061357; doi: 10.1093/rb/rbaa006.

Reply: Thank you for your comment and these useful references. We have added a paragraph “Biomaterials with osteoimmunomodulatory properties can positively modulate immune cells behavior and promote favorable tissue responses during bone regeneration. Numerous methods have been utilized to alter the interaction with immune cells, including the adjustment of chemical or structural properties and the integration of bioactive substances¹⁵. The vital role of immune cells in regulating the function of bone cells makes the paradigm shift of bone biomaterial design to osteoimmunomodulation¹⁶. Macrophages play a crucial role in bone tissue regeneration by regulating immune response, promoting angiogenesis, and modulating the activity of osteoblasts. Their close interaction with bone cells influences bone remodeling and healing. Osteal macrophages (OsteoMacs) represent a distinct subset of macrophages found within skeletal structures. Intriguing discoveries from foundational studies have highlighted their significant contributions to bone physiology, showcasing their pivotal involvement in both bone formation and remodeling processes^{17, 18}.” in the Introduction section in the revised manuscript (*page 3, line 27-40*) with above references.

Comment No.2: In terms of methodology, even though the authors have employed a broad range of techniques in their investigations, the information regarding the used protocols is insufficient and presents major lacunas, therefore making it harder for other researchers to follow and apply the same methodology in their own studies. With this in mind, for the results to be reproduced and to assure a full transparency, I would recommend that the whole methodology section should be rethought and rewritten, offering supplementary information. For example, the Live&Dead staining, cytoskeleton labelling and osteogenic differentiation investigations (ALP and ARS staining) should be submitted to various modifications where either a step by step protocol or the staining solution names should be offered. If the protocol has been explained in detail in a previously published article, a reference could also be provided in its stead. Moreover, I would recommend that the in vitro biological investigations to

be clearly separated into direct contact studies, where the cells have been seeded directly on to the surface of the analysed samples, and indirect contact studies, where extracts and conditioned medium were used. In this way, the methodology will be easier to understand and follow since it will have a logical flow in the experimental steps. A modification in sequence should also be applied to the results section where the in vivo studies should follow the in vitro studies.

Reply: Thank you for your and suggestions. According to your comment, we have rethought and rewritten the experimental methods section to include the Live&Dead staining, cytoskeleton labelling and osteogenic differentiation investigations (ALP and ARS staining) and others. Herein, we have described the step-by-step protocol in detail and added the names of the corresponding staining solutions. In addition, as you suggested, in the Methods section, we have clearly divided the in vitro biological studies into a Direct contact in vitro biological studies section and an indirect contact in vitro biological studies section. You can find these changes in *Page 19 Line 4-8, 21-26, 36-42; Line 8-13; Page 20 Line 20-25, 36; Page 2 Line 7-21*.

Comment No.3: In the results section, the quality of the presented data is quite good and the obtained results are accurately interpreted, however with some irregularities. For example, for both cell types the cytoskeleton fluorescence images are not very conclusive (Figure 2 and Figure 4), therefore discussion regarding cell shape and actin organization on the aforementioned figures is very hard. I would recommend that the authors should replace the images with appropriate ones or if not possible arrows pointing to said structures, should be added. The same observation for the SEM images, where arrows pointing towards the cells should be added. Moreover, the figures legends are too simple, and since the article is intended for a broad range of individuals, a more detailed description is required.

In addition, even though the statistical analysis is accurate, the error bars for each graphic are not defined in the corresponding legends, therefore I recommend that the legends should also be modified accordingly in order to include a detailed definition of the error bars for each graphic.

Reply: Thank you for your comments. In 3D space, cells adhere directly to the surface of the porous scaffold. However, conventional 2D scanning modes can only capture cell morphology within a single plane. Therefore, we configured the Z-axis scanning mode of the confocal microscope (Dragonfly 200) to capture 3D fluorescence images. The Z-axis scanning mode reconstructs each consecutively captured 2D image into a 3D image within a specified spatial height. However, the reconstructed 3D image may not display cell morphology details as effectively as the 2D image (refer to the figure below). Additionally, we analyzed cell morphology on the porous scaffold surface using SEM images. To enhance clarity, we added arrow indications and revised the legends in Figure 4. Furthermore, we redesigned the box-and-whisker plots to illustrate statistical data more effectively. Detailed descriptions of each graph have been included in the legend.

Comment No.4: Even though the conclusive remarks and the data interpretation are valid and reliable, a lacking in the comparative analysis is observed in the discussion section, therefore a more extensive comparison with the already published literature should be added in order to better highlight the novel aspects and advantages of the proposed Zn-based structures.

Reply: Thank you for your professional suggestion. We have thoroughly reviewed the research progress on biodegradable 3D printed Zn scaffolds for bone repair, particularly

focusing on animal studies (*Adv. Sci.* 2023, 2307329, *Bioact. Mater.* 19 (2023) 12–23, *Adv. Sci.* 2023, 2302702, *Acta Biomater.* 145 (2022) 403–415, etc.). However, performing a meaningful comparative analysis between our study and previous ones is challenging due to differences in scaffold design, surface treatment, printing technology, characterization methods, and analysis methods. Hence, we systematically compare the variables of composition, surface, and structure separately within the same experiment. To better highlight the novelty of the Zn-based scaffolds, we also benchmark the materials currently utilized in clinical settings for bone defect repair.

Zn-Li alloys demonstrate superior comprehensive mechanical properties compared to medical-grade pure Ti (see table below). To compare the osteogenesis performance between 3D printed Zn-Li scaffolds and Ti scaffolds, a critical bone defect was created in rabbit femurs (see figure below). At 2 months, the newly formed cortical bone volume was significantly higher in the Zn-Li scaffold than in the Ti scaffold, as evidenced by Micro-CT and histology. Compared to pure Ti scaffold, new bone tissue almost refilled the defect region in the Zn-Li scaffold while cortical bone on lateral sides of the defect was obviously thickened. To compare the osteogenesis performance between 3D printed Zn-Li scaffolds and Ti scaffolds, a critical bone defect was created in rabbit femurs (see figure below). At 2 months, the newly formed cortical bone volume was significantly higher in the Zn-Li scaffold than in the Ti scaffold, as evidenced by Micro-CT and histology. Compared to pure Ti scaffold, new bone tissue almost refilled the defect region in the Zn-Li scaffold while cortical bone on lateral sides of the defect was obviously thickened.

Comparison of key properties between pure Ti, autologous bone and Zn-Li alloys

Materials	Mechanical properties				Biodegradability	Bioactivity	Printability
	UTS ^b (MPa)	UCS ^c (MPa)	Elongation %	Elastic Modulus (GPa)			
Cortical bone	50-151	130-200	-	7-30	No	Yes	No
Pure Ti (Grade 1-4) ^a	240-550	-	15-24	110	No	No	Yes
Zn-Li alloys	252-780	790-1100 ^{d6}	0-26	100	Yes	Yes	Yes

^aASTM-F67

^bUltimate tensile strength

^cUltimate compressive strength

^d Zn-Li alloys have compression super plasticity, the maximum stress before 50% compressive strain was defined as ultimate compressive strength

Figure Bone regeneration comparison between Zn-Li scaffold and Ti scaffold in a critical bone defect rabbit model at 2 months. A Micro-CT reconstruction of new bone tissue and metallic implants with quantitative analysis of bone volume/tissue volume (BV/TV), bone mineral density (BMD), trabecular thickness (Tb. Th), and trabecular separation (Tb. Sp). New bone is marked in yellow, implants are marked in white. B Methylene blue acid fuchsin staining of bone defect regions. Yellow asterisks indicate newly formed bone, white asterisks are scaffold struts. Data are presented as mean \pm standard deviation. P-values are calculated using one-way ANOVA with Tukey's post hoc test, * $p < 0.05$, ** $p < 0.01$, *** $p < 0.005$.

We have added the above content into the Discussion section (*page 13, line 22 to 42, and page 14, line 1 to 12*) in the revised manuscript, the table and figure are added into the Supplementary Information, please check.

Comment No.5: For suggested improvements that could help strengthening the work presented in the current manuscript I would recommend the following:

a) A quantitative test such as MTT or CCK-8 in order to assess the cytotoxic effect of both the alloy and its extract since the Live&Dead analysis is a qualitative viability test

that offers a limited view on the cellular viability and proliferative processes. In addition, the fluorescence images offered are not very conclusive.

Reply: Thanks for your comment. The cytotoxicity of Zn-Li alloy system has been systematically investigated in our published study (*Nat. Comm.* 11, (2020) 401). Zn-0.1-0.8Li alloy extracts (100% or 50%) showed good biocompatibility, and promote the proliferation of MC3T3-E1 cells (see figure below). That's why we use Zn-Li alloy system for 3D printed scaffold in the present work.

Cytocompatibility of Zn-Li alloy extracts on MC3T3-E1 cells

b) A negative control for the macrophage polarization study since from what it can be seen from Figure 8, the cells expressed positive signals for both phenotype markers. Thus in order to eliminate the idea of auto-fluorescence or non-specific labelling, a negative control should be provided for comparison. Another suggestions could include the use of a double labelling or the replacement of a less sensitive marker.

Reply: Thank you for your professional suggestion. Macrophage polarization is a dynamic and complex process characterized by crossover factors and signaling pathways, resulting in macrophages exhibiting a mixed phenotype expressing both M1 and M2-type markers. Notably, reported studies (*Sci. Adv.* 7, eabf6654(2021), *Adv. Funct. Mater.* 2023, 2213128) have demonstrated that even in the resting M0 state, macrophages express both iNOS (M1) and CD206 (M2) markers. The polarization tendency of macrophages towards M1 or M2 is discerned through changes in fluorescence intensity of these markers. Therefore, immunofluorescence staining of polarization markers serves to qualitatively describe the polarization status of macrophages.

Furthermore, we quantified the immunomodulatory factors released by M1 and M2-type macrophages via qRT-PCR. As depicted in Figure 8D, the expression of M1-type macrophage factors (iNos, Il-1 β , Tnf- α) was downregulated by approximately 50% in both the BCC and G scaffold groups, while the expression of M2-type macrophage factors (Il-10, Arg1) was upregulated by approximately 2-fold compared to the control group. Therefore, it is reasonable to infer that the zinc-based porous scaffold extract efficiently promotes macrophage polarization towards the M2-type while inhibiting polarization towards the M1-type.

(*Adv. Funct. Mater.* 2023, 2213128)

(*Sci. Adv.* 7, (2021) eabf6654)

c) In conclusion, the present study is very complex and the concluding remarks are very interesting and quite significant in the context of developing implantable biomaterials with immunomodulatory properties for a proper bone tissue regeneration. It is very clear that the authors belong to a research group with a significant expertise in this domain, therefore the research could be relevant and interesting for various people such as PhD students, young researchers or interdisciplinary teams starting their work in the osteoimmunology field with bone implantable biomaterials for regenerative applications.

Reply: Many thanks for your professional comments to improve the quality of our work.

Reviewer #2 (Remarks to the Author):

Reviewer #3 (Remarks to the Author):

This work presents an innovative and comprehensive approach to addressing the intricate challenges associated with balancing degradation and toxicity in biodegradable materials, particularly in the context of porous scaffolds for bone tissue engineering. The research approach is logically sound, the data are comprehensive and complete, and the argumentation is solid. Nevertheless, there are improvements that should be made to enhance the quality of the current manuscript.

Comment No.1: The study effectively introduces the research focus on balancing degradation and toxicity in Zn-based porous scaffolds through osteoimmunomodulation. However, it would be beneficial to provide more context regarding the significance of this conundrum and its relevance to current challenges in tissue engineering.

Reply: Thanks for your suggestion. Biomaterial scaffold plays a key role as the platform to carry cells and factors (biophysical and chemical factors, peptides, stimuli, etc.) in bone tissue engineering. In the past decades, there is a clear paradigm shift from bioinert scaffold to bioactive or degradable scaffolds (*Nat. Rev. Mater.* 5 (2020) 584-603). Biodegradable metals (Mg, Fe, Zn, etc.), synthetic polymers (PLA, PLGA, PLLA, PPF, etc.), and bioactive ceramics (TCP, CaSO₄, akermanite, etc.) are representative biomaterials in the major categories. Nutrient elements (Ca, Mg, Zn, Sr, Fe, Cu, etc.) are widely used as scaffold materials or additives to promote bone regeneration and repair. The biological function of these elements depends on their concentration, leading to dose-dependent effects (*J. Trace Elem. Med. Biol.* 32 (2015) 86-106). Both too low and too high doses can result in no or negative effects, with only appropriate doses exhibiting beneficial functions. For instance, 13 µg/mL of zinc in cell medium is toxic to pre-osteoblast cells (MC3T3-E1), while 6 µg/mL of zinc promotes cell proliferation (*Nat. Comm.* 11, (2020) 401). The design of bone scaffolds, including parameters such as pore unit, pore size, porosity, etc., undoubtedly influences the degradation dynamics of scaffold materials. Thus, striking a balance between preventing overdose toxic effects and leveraging the beneficial functions of biodegradable materials is critically important in bone tissue engineering. We have incorporated relevant content into the revised manuscript on *page 3, lines 20-25*.

Comment No.2: The authors mention that rapid degradation and excessive toxicity pose a dilemma for biodegradable materials when transitioning from bulk to porous forms. However, they do not demonstrate the difference in degradation behavior and biocompatibility between bulk Zn-Li alloy and Zn-Li alloy scaffolds. I suggest providing more data to support this assertion.

Reply: Thank you for your comment. Please refer to the figure below for a comparison between bulk Zn-Li alloy and Zn-Li alloy scaffolds. As depicted, there is a ring of new bone tissue surrounding the bulk Zn-Li alloy implant at the 3-month mark, indicating excellent osteogenic capability. Additionally, the micro-CT and SEM cross-sectional images illustrate minimal degradation of the bulk sample. In contrast, the BCC scaffold exhibits significant degradation (highlighted by red asterisks), resulting in limited bone regeneration at the 3-month point. The G scaffold demonstrates an appropriate

degradation rate, which falls in between the other two, facilitating bone formation and ingrowth into the pore regions. We have included this result in the Supplementary Information (Figure S4); please review it at your convenience.

Comparison of new bone regeneration and degradation between Zn-Li alloy bulk sample and scaffolds at 3 months. Red asterisks indicate scaffold degradation.

Comment No.3: To achieve a biodegradable scaffold with immunomodulatory capabilities, the authors propose a design strategy that includes chemical composition, surface treatment, and geometry. How do you rank the importance of these factors, and why?

Reply: Thank you for your question. Compositional design serves as a fundamental platform in materials science, governing macro-scale regulation that defines material properties and bioactivity. It acts as a prerequisite for subsequent, more precise modulation. Conversely, surface treatments target cellular and subcellular scale regulation. Manipulating surface roughness enables more precise control over macrophage adhesion and skeleton stretching.

In the case of degradable materials, the nanoscale surface pattern collapses as the material degrades and diffuses in vivo, making surface treatment essential for short-term regulation during the initial stages of bone repair. Material composition and geometry, on the other hand, entail long-term regulation. Geometric design dictates the form of active products. The pore geometry of the scaffold significantly impacts the

diffusion behavior of zinc ions, with variations in diffusion behavior influencing the concentration of locally formed solid products and diffused zinc ions.

These solid products occupy the pore region, affecting bone growth, while the concentration of released zinc ions directly coordinates the inflammatory response and subsequent bone regeneration. Hence, material composition, surface treatment, and geometric design are all essential for precisely regulating bone immunity.

Comment No.4: In Figure 2, the surface patterns between acid etching and EC polishing are quite similar in terms of morphology and roughness, but results of macrophage attachment are markedly different. Could you please provide further explanation of this phenomenon?

Reply: Thank you for your question. As depicted in Fig. 2A and C, the surfaces of the scaffolds following ultrasonic treatment exhibited a disordered nanoscale morphology. However, micrometer-sized unmelted powder and a dense oxide layer still covered the scaffolds. Acid etching partially removed loosely attached powder, yet some unfused spherical powder persisted. Despite revealing microscopic surface patterns, achieving adhesion for the micron-sized RAW264.7 remained challenging. In contrast, the surface of the EC-polished scaffold strut appeared smooth, free from attached powders. Simultaneously, the micro-surface displayed protruding structures and a wavy morphology. These two levels of surface morphology contributed to a more favorable environment for adhesion and cell spreading, particularly for RAW264.7 cells.

Surface patterns and cell morphology

Comment No.5: In many published studies, zinc exhibits cytotoxicity on osteoblast cells. In Figure 4, MC3T3-E1 cells were well attached to the scaffold struts. What factors do you believe contribute to this difference?

Reply: Thank you for your question. Many previous studies examining zinc toxicity to osteoblasts relied on conventional 2D planar cultures. Similarly, in our study, as illustrated in the figure below, we observed that MC3T3-E1 cells cultured on bulk samples for 48 hours exhibited a large number of dead cells adhering to the surface, displaying a crumpled ellipsoid shape with the cytoskeleton not obviously stretched longitudinally. Consequently, excessive doses of zinc did induce toxicity to MC3T3-E1 cells in 2D planar cultures.

Live/dead staining, SEM images and F-actin staining of MC3T3-E1 cells on the bulk sample at 48 h.

However, Fig. 4 reveals MC3T3-E1 cells directly seeded on 3D porous scaffolds, wherein cell behavior was influenced by geometric morphology and surface curvature compared to 2D planar cultures. Gaussian curvature calculations of natural bone trabeculae reveal hyperbolic structures with varying curvatures on the surface (*Bone*, 2002, 30(1): 191-194). The porous scaffold surface closely mimics the curvature of bone trabeculae in vivo, featuring concave and convex curvatures. Studies have shown that cells prefer to adhere and proliferate on concave surfaces compared to planar ones

(*Biomaterials Science*, 2015, 3(2): 231-245), while cells on convex surfaces undergo directed migration through cytoskeletal contraction.

Hence, alterations in the surface curvature of porous scaffolds may significantly impact the attachment rate and morphology of MC3T3-E1 cells, thereby regulating and guiding cell and tissue regeneration (*PNAS*, 2022, Vol. 119 No. 41 e2206684119; *Nat. Commun.* (2023) 14, 855). We appreciate the reviewer's question, as it enabled us to directly seed MC3T3-E1 cells on zinc-based porous scaffolds with varying geometrical shapes for the first time. Understanding why MC3T3-E1 cells can adhere well to scaffold struts is a key aspect of our research. Moving forward, our next step will be to further investigate the effects of geometrical shape and surface curvature on cell behavior.

Comment No.6: The sample size (n) in Figure 3A, 4C, 6, and 8 should be included in the figure legend. Gene names mentioned in this study require thorough checking and should appear in italics.

Reply: Thank you for your comment. We have added the sample size (n) in the corresponding figure legend. We have checked and corrected the gene names in our revised manuscript and have used italics for them. Please check.

Comment No.7: It is intriguing that scaffolds with different geometries (G and BCC) degrade in distinct manners. I suggest the authors discuss this aspect in more detail as it will illuminate how to properly design degradable scaffolds for other researchers.

Reply: Thank you for your question. Zn-based scaffolds undergo degradation upon contact with body fluids, and structural parameters, such as specific surface area (SSA), play a crucial role in determining the initial degradation rate. In general, a larger SSA corresponds to a faster initial degradation rate. The surface geometry also influences the degradation mode of the scaffold. The G sample exhibits small and shallow pits distributed homogeneously in the double-curved struts, while the BCC sample displays a non-uniform mode with large and deep pits concentrated in the connection region of struts. Moreover, the diffusion coefficient and its anisotropy may impact the degradation behavior of scaffolds. Diffusion tests reveal that the BCC unit demonstrates a strong anisotropy in terms of diffusion coefficient compared to the G unit, with the maximal diffusion coefficient of the BCC scaffold being 1.6 times larger than that of

the G scaffold. Consequently, the BCC scaffold exhibits a faster degradation rate and a more non-uniform degradation mode. We have incorporated this information into the results section (*page 6, lines 32-35; page 7, lines 10-13*) and the discussion section (*page 12, lines 26-31, 36-42*) of the revised manuscript.

Comment No.8: There is a notable disparity in the amount of newly formed bone observed inside the pores of the G scaffold compared to the nearly absent bone tissue found in the BCC scaffold, which presents a stark contrast. What factors contribute to this outcome?

Reply: Thank you for your question. Compared to G scaffolds, BCC scaffolds exhibited greater anisotropy in diffusion behavior. At 1 month, nearly half of the BCC scaffold was occupied by inhomogeneous solid degradation products within the pore units. In contrast, the G scaffold degraded uniformly, leaving most of the pore geometry available for tissue growth (refer to Figure 3C). Consequently, degradation products, including ion release and distribution, coordinated the inflammatory response at the interface of the pore unit, leading to differences in bone growth into BCC and G scaffolds.

Three days after implantation, M1-type macrophages were predominantly distributed at the BCC scaffold interface, whereas the G scaffold had an M2/M1 ratio of approximately 1:1. By 1 month, the M2/M1 ratio of G scaffolds was more than twice that of BCC scaffolds (see Figure 6). This suggests that the direction and degree of macrophage polarization from the early stages of the inflammatory response determine the difference in bone immunomodulatory capacity between BCC and G scaffolds.

Consequently, at 1 month, early osteogenic factor expression (ALP and OCN) was predominantly found in G scaffolds, whereas at 3 months, CT and histology demonstrated superior bone growth into G scaffolds. This corroborates findings from previous studies (*Biomaterials* 276 (2021) 121037; *Bioactive Materials* 6 (2021) 757–769; *Acta Biomaterialia* 156 (2023) 222–233), indicating that inconsistencies in the direction and extent of bone immunoregulation result in differences in bone regeneration and ingrowth. Furthermore, we have provided a more detailed explanation in the Discussion section (refer to *page 12, lines 23-42; page 13, lines 1-30*).

Comment No.9: It appears that when Zn-based bone scaffolds degrade rapidly, they may interfere with regular bone repair and lead to the failure of bone healing. How can this be prevented from happening from a clinical translation perspective?

Reply: Thank you for your question. One of the main objectives of this study is to assess the feasibility of utilizing biodegradable Zn scaffolds for bone repair. That's why we 3D printed scaffolds with 90% porosity, the maximum achievable with this technology. As you can observe, despite such high porosity, careful modulation of the chemical composition, surface treatment, and pore geometry contributes to a sophisticated immunomodulation mechanism, resulting in robust bone regeneration and ingrowth into the Zn-Li scaffolds.

In a clinical setting, the porosity would certainly be reduced to enhance the load-bearing properties of the scaffolds, with porosities ranging from 40-60% being more practical. Consequently, the degradation rate and corresponding ion release would decrease with reduced porosity, thereby minimizing the risk of side effects. Importantly, our work demonstrates the feasibility of achieving Zn-based porous scaffolds with enhanced osteogenesis through the deliberate design of scaffold parameters such as pore size, porosity, specific surface area, anisotropy, diffusion coefficient, and so forth.

Comment No.10: While the study highlights the enhanced efficiency of the G scaffold in promoting anti-inflammatory macrophage polarization and osteogenesis, it would be beneficial to discuss the potential clinical applications of this technology and outline future research directions aimed at optimizing scaffold design and performance for specific tissue engineering applications.

Reply: The Zn-Li bone scaffold studied here offers a unique advantage due to its combination of high strength, biodegradability, bioactivity, and printability. Consequently, its potential clinical applications are targeted toward bone defects at load-bearing sites, such as critical segmental defects in long bones. Current treatments for such bone defects typically involve autologous cortical bone grafts or metallic implants, particularly 3D printed titanium scaffolds¹. While autologous bone grafting is considered the gold standard, it requires an additional surgical procedure and is often associated with donor site morbidity or insufficient graft material. Titanium exhibits excellent mechanical performance and biocompatibility, facilitating good osseointegration with new bone tissue. Utilizing 3D printing technology, titanium scaffolds can be customized to meet patients' individual needs for bone implants. However, titanium scaffolds remain in place permanently after new bone regeneration and ingrowth, leading to long-term interference with bone remodeling and potential chronic inflammation due to trace Ti ion release. Moreover, titanium scaffolds are not ideal for treating bone defects in complex situations such as osteoporosis or bone infections due to their bioinert nature.

Zn-Li alloys demonstrate superior comprehensive mechanical properties compared to medical-grade pure Ti (see table below). In a rabbit tibial fracture model, the Zn-Li alloy screw and plate system exhibited comparable performance to Ti-6Al-4V alloy counterparts[2]. Furthermore, it is possible to modulate bioactivities through alloying with functional elements. For instance, alloying with trace amounts of Sr resulted in a Zn-0.8Li-0.1Sr intramedullary nail showing superior osteogenesis-inducing and osteoporotic-bone-fracture-treating effects compared to pure Ti[3]. Zn-Li-Ag demonstrated potent bactericidal effects against methicillin-resistant Staphylococcus aureus (MRSA), leading to remarkable infection control and favorable bone retention in an MRSA-induced rat femoral osteomyelitis model[4]. To compare the osteogenesis performance between 3D printed Zn-Li scaffolds and Ti scaffolds, a critical bone defect was created in rabbit femurs (see figure below). At 2 months, the newly formed cortical bone volume was significantly higher in the Zn-Li scaffold than in the Ti scaffold, as evidenced by Micro-CT and histology. Compared to pure Ti scaffold, new bone tissue almost refilled the defect region in the Zn-Li scaffold while cortical bone on lateral sides of the defect was obviously thickened. The future research topic will be focused on building the quantitative logic relationship between scaffold structural parameters and key properties such as degradation rate and mode, strength, and bone growth, and multi-performance collaborative optimization.

Comparison of key properties between pure Ti, autologous bone and Zn-Li alloys

| Materials | Mechanical properties | | | | Biodegradability | Bioactivity | Printability |
|---|---|---|---|---|---|---|---|
| | UTS[b] (MPa) | UCS[c] (MPa) | Elongation % | Elastic Modulus (GPa) | | | |

Cortical bone ⁵	50-151	130-200	-	7-30	No	Yes	No
Pure Ti (Grade 1-4) ^a	240-550	-	15-24	110	No	No	Yes
Zn-Li alloys	252-780	790-1100 ^{d6}	0-26	100	Yes	Yes	Yes

^aASTM-F67

^bUltimate tensile strength

^cUltimate compressive strength

^dZn-Li alloys have compression super plasticity, the maximum stress before 50% compressive strain was defined as ultimate compressive strength

Figure Bone regeneration comparison between Zn-Li scaffold and Ti scaffold in a critical bone defect rabbit model at 2 months. A Micro-CT reconstruction of new bone tissue and metallic implants with quantitative analysis of bone volume/tissue volume (BV/TV), bone mineral density (BMD), trabecular thickness (Tb. Th), and trabecular separation (Tb. Sp). New bone is marked in yellow, implants are marked in white. B Methylene blue acid fuchsin staining of bone defect regions. Yellow asterisks indicate newly formed bone, white asterisks are scaffold struts. Data are presented as mean \pm standard deviation. P-values are calculated using one-way ANOVA with Tukey's post hoc test, * $p < 0.05$, ** $p < 0.01$, *** $p < 0.005$.

References

1. Pobloth, A. M., Checa. S., Razi, H., et al. Mechanobiologically optimized 3D titanium-mesh scaffolds enhance bone regeneration in critical segmental defects in sheep. *Sci. Transl. Med.* **10** 423 (2018).

2. Yang, H., Qu, X., Wang, M., et al. Zn-0.4 Li alloy shows great potential for the fixation and healing of bone fractures at load-bearing sites. *Che. Eng. J.* **417** 129317 (2021).
3. Jia, B., Zhang, Z., Zhuang, Y., et al. High-strength biodegradable zinc alloy implants with antibacterial and osteogenic properties for the treatment of MRSA-induced rat osteomyelitis. *Biomaterials* **287** 121663 (2022).
4. Zhang, Z., Jia, B., Yang, H., et al. Zn-0.8Li-0.1Sr-a biodegradable metal with high mechanical strength comparable to pure Ti for the treatment of osteoporotic bone fractures: In vitro and in vivo studies. *Biomaterials* **275** 120905 (2021).
5. Gerhardt, L. C., Boccaccini, A. R. Bioactive glass and glass-ceramic scaffolds for bone tissue engineering. *Materials* **3**, 3867-3910 (2010).
6. Yang, H., Jia, B., Zhang, Z., Qu, X., Li, G., Lin, W., Zhu, D., Dai, K., Zheng, Y. Alloying design of biodegradable zinc as promising bone implants for load-bearing applications. *Nat. commun.* **11**, 1-16 (2020).

Reviewers' Comments:

Reviewer #1:

Remarks to the Author:

After following the offered recommendations and by providing extended and informative answers to all of the raised questions, the quality of the submitted manuscript has been significantly improved in comparison to its previous form. However, I still think that certain issues need to be addressed and that several modifications are still required ahead of publication, in order to meet the high quality criteria of the Nature Communications journal.

Comment No.1: Although the authors have provided detailed step by step protocols for the experimental approach, I would recommend a revision of the newly added paragraphs, since in the present form they are really hard to follow due to writing mistakes (e.g. "ALP staining were performed"; "previous description"; "conditioned medium were prepared", etc.) and lack of a specific phraseology (e.g. "drained for fluorescence imaging"; "the multiwell scaffolds were directly placed into the working solution"; etc). Moreover, please remove the following sentence: "Follow the instructions for the experiment" found at pp.20, line 21.

Comment no.2: In order to avoid repetition please remove the "Cell culture" subsection found at pp. 18, lines 25-35.

Comment no.3: In order to be constant, please use either the full or shorter version of "minutes" and "hours", since throughout the whole length of the manuscript both forms have been used.

Comment no.4: As stated before, the "In vitro macrophage cytokines modulate osteogenic and differentiation of MC3T3-E1" section should be moved ahead of the in vivo study, in order to have a logical flow of the experimental approach.

In conclusion, I consider that the present manuscript could bring an important contribution to the osteoimmunology field, but only if the raised points would be addressed properly.

Reviewer #2:

Remarks to the Author:

Reviewer #3:

Remarks to the Author:

The authors have addressed the reviewers' concerns and this reviewer recommends to accept for publication in Nature Communications.

Response to Reviewer

Reviewer #1 (Remarks to the Author):

After following the offered recommendations and by providing extended and informative answers to all of the raised questions, the quality of the submitted manuscript has been significantly improved in comparison to its previous form. However, I still think that certain issues need to be addressed and that several modifications are still required ahead of publication, in order to meet the high quality criteria of the Nature Communications journal.

Comment No.1: Although the authors have provided detailed step by step protocols for the experimental approach, I would recommend a revision of the newly added paragraphs, since in the present form they are really hard to follow due to writing mistakes (e.g. “ALP staining were performed”; “previous description”; “conditioned medium were prepared”, etc.) and lack of a specific phraseology (e.g. “drained for fluorescence imaging”; “the multiwell scaffolds were directly placed into the working solution”; etc). Moreover, please remove the following sentence: “Follow the instructions for the experiment” found at pp.20, line 21.

Reply: Thank you for your comments. We have revised the added paragraphs to fix writing errors and use more specific phraseology. You can find these changes in Page 18 Line 31-39; Page 19 Line 9-13; Page 20 Line 5-10, 16-19, 30-37. In addition, as you suggested, we have removed the “Follow the instructions for the experiment”. Please check.

Comment no.2: In order to avoid repetition please remove the “Cell culture” subsection found at pp. 18, lines 25-35.

Reply: Thank you for your comments. We have removed the “Cell culture” subsection.

Comment no.3: In order to be constant, please use either the full or shorter version of “minutes” and “hours”, since throughout the whole length of the manuscript both forms have been used.

Reply: Thank you for your comments. We have corrected the “minutes” and “hours” in the whole manuscript to shorter versions of “min” and “h”.

Comment no.4: As stated before, the “In vitro macrophage cytokines modulate osteogenic and differentiation of MC3T3-E1” section should be moved ahead of the in vivo study, in order to have a logical flow of the experimental approach.

In conclusion, I consider that the present manuscript could bring an important contribution to the osteoimmunology field, but only if the raised points would be addressed properly.

Reply: Thank you very much for your suggestion. we have moved the section of “In vitro macrophage cytokines modulate osteogenic and differentiation of MC3T3-E1” ahead of the in vivo study in the revised manuscript, please check.

Reviewer #2 (Remarks to the Author):

Reviewer #3 (Remarks to the Author):

The authors have addressed the reviewers' concerns and this reviewer recommends to accept for publication in Nature Communications.

Reply: We thank the reviewer for the positive response and all valuable suggestions in previous comments to help us improve our manuscript.